

# Towards the closure of momentum budget analyses in the WRF (v3.8.1) model

Ting-Chen Chen, Man-Kong Yau and Daniel J. Kirshbaum

Atmospheric and Oceanic Sciences Department, McGill University, Montreal, H3A0B9, Canada

*Correspondence to*: Ting-Chen Chen (ting-chen.chen@mail.mcgill.ca)

**Abstract.** Budget analysis of a tendency equation is widely utilized in numerical studies to quantify different physical processes in a simulated system. While such analysis is often post-processed when the output is made available, it is well-acknowledged that the closure of a budget is difficult to achieve without averaging. Nevertheless, the potential rise of the errors in such calculation has not been systematically investigated. In this study, an inline budget retrieval method is first

developed in the WRF v3.8.1 model and tested on a 2D idealized slantwise convection case with a focus on the momentum equations. This method extracts all the budget terms following the model solver, which gives a high accuracy with a residual term always less than 0.02% of the tendency term. Then, taking the inline values as truth, several post-processing budget analyses with different commonly-used simplifications are performed to investigate how they may affect the accuracy of the estimation of individual terms and the resultant residual. These assumptions include using a lower order advection operator

than the one used in the model, neglecting the C staggering grids, or following a mathematically-equivalent but transformed format of equation. Errors in these post-processed analyses are found mostly over the area where the dynamics are the most active, impairing the subsequent physical interpretation. A maximum 99th percentile residual can reach 800% of the concurrent tendency term, indicating the danger of neglecting the residual term as done in many budget studies. This work provides general guidance not only for applying an inline budget retrieval to the WRF model but for minimizing the errors in

post-processing budget calculations.

## 1 Introduction

The atmosphere is a complex system with different scales of motion. Its dynamics are governed by a set of fluid equations based on the fundamental laws of physics. Although the equation set cannot be solved analytically, numerical models can be used to simulate the observed weather and climate systems to improve our understanding of the atmosphere.

Due to the complexity and nonlinearity of the numerical models, budget analysis is often employed to interpret the results by quantifying the contribution of each term (i.e., physical process) in a tendency equation that governs the evolution of a certain quantity in the simulated system. The accuracy of budget analysis can be estimated from the residual term, defined as the difference between the tendency term on the left-hand side (lhs) of the equation and the summation of all the forcing terms on its right-hand side (rhs). Budget analysis has been performed on diverse properties (e.g., momentum, temperature,



water vapor, vorticity, etc) of many systems on various scales, including the Madden-Julian oscillation (MJO; e.g., Kiranmayi and Maloney, 2011; Andersen and Kuang, 2012), tropical cyclones (e.g., Zhang et al., 2000: Rios-Berrios et al., 2016; Huang et al., 2018), squall lines (e.g., Sanders and Emanuel, 1977; Gallus and Johnson, 1991; Trier et al., 1998), supercell thunderstorms (e.g., Lily and Jewett, 1990) and so on.

Despite the popularity of the budget analysis, it is generally acknowledged that, in model post-processing analysis,
obtaining a closed budget with a negligible residual is difficult (e.g., Kanamitsu and Saha, 1996) and has been accomplished mostly in time or domain averaged budget calculations (e.g., Lilly and Jewett, 1990; Balasubram and Yau, 1994; Arnault et al., 2016; Kirshbaum et al., 2018; Duran and Molinari, 2019). Even in the case of averaged budgets, the residual term that contains non-explicitly-diagnosed physics can be larger than the tendency term (e.g., Liu et al., 2016) and many studies simply do not display the residual, making the proper interpretation of the budget analysis difficult.

The "residual analysis method" is sometimes utilized to obtain an indirect estimation of the physical processes that are hard to diagnose or are unresolved in a set of analysis data. In such cases, a non-negligible residual is sometimes used to gain insight into such processes. However, as just discussed, the residual term also contains the inaccuracies associated with the calculations within the budget analysis (e.g., Kornegay and Vincent, 1976; Abarca and Montgomery, 2013). It is thus unclear whether the unsolved physics indeed comprise of the main component of the residual without considering the contribution of
other sources of errors in the budget calculation (Kuo and Anthes, 1984). Whereas it is almost impossible to separate the subgrid-scale, unsolved processes from other errors in reanalysis or observational data (e.g., Hodur and Fein, 1977; Lee 1984), the focus of this study is on numerical model data where the local tendency and all the associated resolved and parameterized physics can be obtained from the model. Thus, the residual term in this study specifically refers to errors in the budget calculation.

To reduce the residual, an inline budget analysis that extracts all the terms of a prognostic equation directly from the model during its integration is generally the most accurate. However, the procedure has been reported only in a few studies (e.g., Zhang et al., 2000; Lehner, 2012; Moisseeva, 2014). Most other studies still conduct the offline/post-processing budget analysis when the output is made available after the model integration. Some specific suggestions have been given in the past regarding how to reduce the error of post-processed budget analysis. For example, Lilly and Jewett (1990) emphasized the
importance of evaluating terms using the same differencing scheme, grid stretching, and grid staggering as that used in the simulation model. However, it is uncertain whether these rules have been widely followed, and how much of a reduction in residual can be obtained with this approach.

In some post-processing budget analyses, transformed equations with different assumptions from those in the model are used and naturally lead to errors in the budget results. On the other hand, even when the same form of the equations is
followed, errors can still arise from multiple sources during the post-processing. Some errors are inherent in the time discretization scheme of the model, some are traced to the numerical methods in solving the temporal/spatial derivatives with finite differencing (e.g., Kuo and Anthes, 1984), and others might emerge during the inter/extrapolation from model grids to analysis grids (e.g., Lilly and Jewett, 1990). While the tendency term is often the result of a few cancellations among





competing forcing terms, the seemingly non-dominant terms may be as important as the large forcing terms in determining
the sign and the value of the tendency. Thus, an incorrect estimation of even a small term may result in a residual with
magnitude comparable to the tendency term, thus hindering the subsequent physical interpretation.

A few models, such as Cloud Model 1 (CM1; Bryan and Fritsch, 2002) and High Resolution Limited Area Model
(HIRLAM; Undén et al., 2002), offer users the choice in the namelist file to output the budget terms for prognostic variables.
However, many other models (e.g., Fifth-Generation NCAR / Penn State Mesoscale Model (MM5; Grell et al., 1994),
Weather Research and Forecasting Model (WRF; Skarmarock et al., 2008), the Advanced Regional Prediction System
(ARPS; Xue et al., 2000, 2001) and the Regional Atmospheric Modeling System (RAMS; Pielke et al., 1992)) do not have
the capability to write out all these terms into the standard history output files. In this study, we develop an inline momentum
budget retrieval method in the Advanced Research WRF model, one of the most widely-used numerical weather prediction
models. During the period 2011-15, there are on average 510 peer-reviewed journal publications involving WRF per year
(Powers et al., 2017). Given the widespread use of WRF for both real-case and idealized modelling, such a tool may prove
useful in numerous applications. In the method, each contributing term is extracted during the model integration and stored
as a standard output. In so doing, we essentially solve the prognostic variables as done in the model and so the two sides of
the tendency equation are always in balance regardless of the output time interval. By taking the results from the inline
budget analysis as truth, we then perform several different post-processing budget analyses with commonly-made
simplifications or a different format of equation. Comparisons between the post-processing budgets and the inline/true values
are made to investigate the potential increase in errors in each forcing term and the resultant residuals.

## 2 Model and numerical setup

### 2.1 Model and momentum equations

The WRF configuration used in this study is a two-dimensional [(y, z); no variation in the x direction], fully
compressible, non-hydrostatic and idealized version of the Advanced Research WRF model, version 3.8.1 (Skamarock et al.,
2008). Here we briefly revisit the parts that are relevant to the momentum budget analysis. The governing equations in the
WRF model are cast on a terrain-following dry-hydrostatic pressure coordinate. This vertical coordinate, η, is defined as

$$\eta = \left(p_{dh} - p_{dh\_top}\right)/\mu_d$$

where $p_{dh}$ is the hydrostatic pressure of the dry air and $\mu_d$ represents the mass of the dry air per unit area in the column,
$\mu_d = p_{dh\_sfc} - p_{dh\_top}$ where $p_{dh\_sfc}$ and $p_{dh\_top}$ indicate the values of $p_{dh}$ at the surface and the top of the dry
atmosphere, respectively.

To ensure conservation properties, the model equations are formulated in flux form, with the prognostic variables
coupled with $\mu_d$. The flux-form momentum components are defined as

$$U = \mu_d u, \qquad V = \mu_d v, \qquad W = \mu_d w, \qquad \Omega = \mu_d \frac{d\eta}{dt},$$



where u, v and w are the zonal, meridional and vertical velocities, respectively. Note that the dry-mass-coupled velocities ($U$, $V$, $W$) on coordinates (x, y, z) have units of Pa m $s^{-1}$, and the dry-mass-coupled vertical velocity on η coordinate, Ω, has a unit of Pa $s^{-1}$. For the idealized 2D case on a $f$-plane as in this study, the momentum equations in the WRF model are written as

$$\underbrace{\frac{\partial V}{\partial t}}_{\substack{V \text{ tendency}}} = \underbrace{-\nabla \cdot (\vec{V}v)}_{\substack{\text{advection} \\ \text{ADV}}} \underbrace{-\mu_d \alpha \frac{\partial p}{\partial y} - \frac{\alpha}{\alpha_d} \frac{\partial p}{\partial \eta} \frac{\partial \phi}{\partial \eta}}_{\substack{\text{horizontal pressure gradient force} \\ \text{PGF}}} \underbrace{-fU}_{\substack{\text{Coriolis} \\ \text{COR}}} \underbrace{-\left(\frac{vW}{r_e}\right)}_{\substack{\text{curvature} \\ \text{CUV}}} + \underbrace{P_V}_{\substack{\text{remaining} \\ \text{(parameterized)} \\ \text{physics}}} + res \qquad (1)$$

$$\underbrace{\frac{\partial W}{\partial t}}_{\substack{W \text{ tendency}}} = \underbrace{-\nabla \cdot (\vec{V}w)}_{\substack{\text{advection} \\ \text{ADV}}} + \underbrace{g\left(\frac{\alpha}{\alpha_d} \frac{\partial p}{\partial \eta} - \mu_d\right)}_{\substack{\text{net vertical pressure gradient} \\ \text{and buoyancy force} \\ \text{PGFBUOY}}} + \underbrace{\left(\frac{uU + vV}{r_e}\right)}_{\substack{\text{curvature} \\ \text{CUV}}} + \underbrace{P_W}_{\substack{\text{remaining} \\ \text{(parameterized)} \\ \text{physics}}} + res \qquad (2)$$

where

$$-\nabla \cdot (\vec{V}a) = -\frac{\partial(Ua)}{\partial x} - \frac{\partial(Va)}{\partial y} - \frac{\partial(\Omega a)}{\partial \eta} \qquad (3)$$

is the flux-form advection, $p$ is the full pressure with inclusion of vapor, $\phi$ is the geopotential, $f$ is the Coriolis parameter, $r_e$ is the mean earth radius, and $\alpha$ and $\alpha_d$ are the full and dry-air specific volume, respectively. In our selected microphysics

scheme (Thompson et al., 2008), six hydrometeors are included, and thus $\alpha = \alpha_d \big(1 + q_v + q_c + q_r + q_i + q_s + q_g\big)^{-1}$, where $q_v, q_c, q_r, q_i, q_s$ and $q_g$ are the mixing ratios for water vapor, cloud, rain, ice, snow, and graupel, respectively. The rhs forcing terms for the V tendency include the flux-form advection (ADV), horizontal pressure gradient force (PGF), Coriolis force (COR), vertical (earth-surface) curvature (CUV) and the remaining physics ($P_V$). For the W tendency, the rhs forcings contain the flux-form advection (ADV), net force between the vertical pressure gradient and buoyancy (PGFBUOY),

curvature effect (CUV) and the remaining physics ($P_W$). The remaining physics may include diffusion, vertical velocity damping processes and other parameterized physics, depending on the model setup. Note that for closing the budget analysis, all the known physics processes that come into play should be explicitly written in the equation and be diagnosed or directly retrieved from the model. The residual ($res$) is added on the last rhs term of Eqs. (1) and (2) to represent the imbalance between the two sides of the equation during budget analysis, but it is not part of the original equations solved in the model.

To develop an inline budget retrieval tool, it is very important to understand how these prognostic variables are advanced in the WRF model. Governing equations are first recast to perturbation forms with respect to a hydrostatically-balanced reference state that is strictly a function of height only (defined at initialization) to reduce truncation errors and machine rounding errors. Specifically, variables of $p, \phi, \alpha_d$ and $\mu_d$ are separated into reference and perturbation components, e.g., $p(x, y, \eta, t) = \bar{p}(z) + p'(x, y, \eta, t)$. The introduction of these perturbation variables only changes the expressions for rhs

terms PGF and PGFBUOY in Eqs. (1) and (2), which will not be shown here for simplicity. Readers can refer to Skamarock et al. (2008, chapter 2.5) for more details.





Adapting from Skamarock et al. (2008), Figure 1 summarizes the WRF integration strategy. The integration is wrapped by a third-order Runge-Kutta (RK3) scheme, in which the prognostic variables (generalized as $\Phi$ here) are advanced from $t$ to $t + \Delta t$ given their corresponding partial time derivative equations, $\frac{\partial \Phi}{\partial t} = F(\Phi)$, following a three-step strategy:


$$\Phi^* = \Phi^t + \frac{\Delta t}{3} F(\Phi^t)$$

$$\Phi^{**} = \Phi^t + \frac{\Delta t}{2} F(\Phi^*)$$

$$\Phi^{t+\Delta t} = \Phi^t + \Delta t F(\Phi^{**}) \tag{4}$$

where $\Delta t$ is the model integration time step and $F$, defined as the large-step forcing, represents the summation of all the rhs terms of Eqs. (1) and (2) excluding the residual. Although the parameterized forcings stay fixed from step one to three as

most of the parameterization schemes are called only once at the first RK3 step, the rest of the non-parametrized forcings and thus the total $F$ are changed with the updated $\Phi^*$ and $\Phi^{**}$ at the second and third RK3 step. Within each RK3 step, a subset of time integration with a relatively smaller time step is embedded to accommodate high-frequency/acoustic modes for numerical stability (Wicker and Skamarock, 2002; Klemp et al., 2007; Skamarock et al., 2008). A maximum number for acoustic steps in one model integration step can be specified by the users. To improve accuracy in the temporal solver, the

variables being advanced in this acoustic-step integration are the temporal perturbation fields, defined by the deviation from their more recent RK3 predictors: $\Phi'' = \Phi - \Phi^{t*}$, where $\Phi^{t*} = \Phi^t$, $\Phi^*$ and $\Phi^{**}$ for the first, second, and third RK3 step, respectively. Thus, the perturbation momentum equations to be solved are driven by the large-step forcings and the acoustic-step corrections:

$$\underbrace{\frac{\partial V''}{\partial t}}_{\substack{\text{V tendency}}} = \left[ \underbrace{-\nabla \cdot (\vec{V} v)}_{\substack{\text{advection} \\ \text{ADV}}} \underbrace{- \mu_d \alpha \frac{\partial p}{\partial y} - \frac{\alpha}{\alpha_d} \frac{\partial p}{\partial \eta} \frac{\partial \phi}{\partial y}}_{\substack{\text{horizontal pressure gradient force} \\ \text{PGF}}} \underbrace{-fU}_{\substack{\text{Coriolis} \\ \text{COR}}} \underbrace{- \left( \frac{vW}{r_e} \right)}_{\substack{\text{curvature} \\ \text{CUV}}} + \underbrace{P_V}_{\substack{\text{remaining} \\ (\text{parameterized}) \\ \text{physics}}} \right]^{t*}}_{\substack{\text{large-step forcings (F)}}}$$


$$\underbrace{- \frac{\alpha^{t*}}{\alpha_d^{t*}} \left[ \mu_d^{t*} \left( \alpha_d^{t*} \frac{\partial p''^\tau}{\partial y} + \alpha_d''^\tau \frac{\partial \bar{p}}{\partial y} + \frac{\partial \phi''^\tau}{\partial y} \right) + \frac{\partial \phi^{t*}}{\partial y} \left( \frac{\partial p''}{\partial \eta} - \mu_d'' \right)^\tau \right]}_{\substack{\text{acoustic-step modes (ACOUS)}}} \tag{5}$$

$$\underbrace{\frac{\partial W''}{\partial t}}_{\substack{W'' \text{ tendency}}} = \left[ \underbrace{-\nabla \cdot (\vec{V} w)}_{\substack{\text{advection} \\ \text{ADV}}} + \underbrace{g \left( \frac{\alpha}{\alpha_d} \frac{\partial p}{\partial \eta} - \mu_d \right)}_{\substack{\text{net vertical pressure gradient} \\ \text{and buoyancy force} \\ \text{PGFBUOY}}} + \underbrace{\left( \frac{uU + vV}{r_e} \right)}_{\substack{\text{curvature} \\ \text{CUV}}} + \underbrace{P_W}_{\substack{\text{remaining} \\ (\text{parameterized}) \\ \text{physics}}} \right]^{t*}}_{\substack{\text{large-step forcings (F)}}}$$





$$+g\overline{\left\{\left(\frac{\alpha^{t*}}{\alpha_d{}^{t*}}\right)\left[\frac{\partial}{\partial\eta}\left(C\frac{\partial\phi''}{\partial\eta}\right)+\frac{\partial}{\partial\eta}\left(\frac{c_s^2}{\alpha^{t*}}\frac{\Theta''}{\Theta^{t*}}\right)\right]-\mu_d''\right\}}^{\tau}$$

$$\underbrace{\phantom{+g\overline{\left\{\left(\frac{\alpha^{t*}}{\alpha_d{}^{t*}}\right)\left[\frac{\partial}{\partial\eta}\left(C\frac{\partial\phi''}{\partial\eta}\right)+\frac{\partial}{\partial\eta}\left(\frac{c_s^2}{\alpha^{t*}}\frac{\Theta''}{\Theta^{t*}}\right)\right]-\mu_d''\right\}}}}_{acoustic-step\ modes\ (ACOUS)}$$

(6)

where $\tau$ indicates the time in the acoustic integration, and $C$ as well as $c_s^2$ are sound-wave related terms (Skamarock et al.,

2008, chapter 3.1.2). Here we leave out the details regarding the acoustic-step terms that are trivial to the inline budget

retrieval. Note that the overbar in Eq. (6) indicates a forward-in-time averaging operator for the acoustic-step modes to damp

instabilities associated with vertically-propagating sound waves (see Eq. (3.19) in Skamarock et al., 2008). Equations (5) and

(6) are the ones used to integrate the prognostic momentum fields in the WRF model. For each RK3 step, after the total

large-step forcing $F$ is determined, V″ and W″ are defined and advanced within the acoustic-step scheme by a loop that adds

F multiplied by a time interval, $\Delta\tau$, (varies with different RK3 steps; see Fig. 1) and the acoustic-step forcing (ACOUS). After

the acoustic-step integration loop ends, V and W are then recovered from their temporal perturbation fields and moved

forward to the next RK3 step. While it is not relevant to the momentum equations discussed here, for some variables

contributed by the microphysics scheme, the associated contribution should be considered after the RK3 integration loop

ends as the microphysics are integrated externally using an additive time splitting (Fig.1) (Skamarock et al., 2008, chapter

3.1.4).

### 2.2 Experimental setup

The main discussion of this study will focus on a 2D (y, z) idealized simulation of slantwise convection. This process

releases conditional symmetric instability (CSI) and assumes no variation along the direction of thermal winds (denoted as

the x direction in our setup; Markowski and Richardson, 2010, chapter 3.4). The initial field consists of a thermally-balanced

uniform westerly wind shear in x. This baroclinic environment contains no conditional (gravitational) instability and no dry

symmetric instability but some CSI. A two-dimensional bubble containing perturbations of potential temperature and zonal

wind is inserted to initiate convection and a slanted secondary circulation (v, w). See the appendix for more details about the

experimental setup. The domain size is 1600 km and 16 km in the y and z direction, respectively, with a horizontal grid

length of 10 km and 128 vertical layers. The model integration time step is 1 minute. For simplicity, only the Thompson

microphysics scheme (Thompson et al. 2008) is used among all the parameterization schemes and the vertical-velocity

damping (Skarmarock et al., 2008, chapter 4.5.1) is also activated. The former does not directly contribute to the momentum

fields and the latter only affects the W momentum budget (contained in $P_W$ in Eqs. (2) and (6)). The WRF model offers

different orders of advection operators, and the default third- and fifth-order operators are selected for the vertical and

horizontal advection in this case, respectively. Most of the analyses and discussion in this paper are based on this slantwise

convection case with a 10-km grid length unless specified otherwise. Two other simulations, one of which uses the same

setup but with an increased horizontal resolution of 2 km will be discussed in Sect. 4.

Figure 2 shows the two-day evolution of the 99th percentiles of v and w and their tendencies. For the 10-km case, the

horizontal velocity reaches its peak in about 20 hours, a few hours after the vertical velocity reaches a maximum, and then





undergoes a weakening. Both v and w tendencies are maximized at around 15 hours. To understand the evolution of the associated flow dynamics, a momentum budget analysis serves as a natural choice. However, as a preliminary step prior to carrying out such analysis, we focus only on the technical discussion of the budget-analysis methodology. The physical interpretation of the motion is beyond the current scope and will be presented in a companion paper.

## 3 Methodology and results

### 3.1 Inline momentum budget analysis


For the inline budget analysis, all the terms are retrieved directly from the model for all the integration time steps, and therefore they represent the "instantaneous" terms that act over the specified short integration time window. For the large-step forcing, the WRF model accumulates all forcing terms at the beginning of each RK3 step. To separate them, we simply take the difference before and after WRF calls the subroutine for each large-step forcing, store their values separately, and

output only the values at the third RK3 step (the total forcing is $F(\Phi^{**})\Delta t$ as shown in Fig. 1). As for the contribution of the acoustic-step modes, they are obtained by accumulating over all the acoustic steps in the third RK3 step ($[ACOUS\ sum]$ shown in Fig. 1). It is worth noting that Eq. (6) is a vertically implicit equation that couples with the geopotential tendency equation (Skarmarock et al. 2008 and Klemp et al. 2007). A tri-diagonal equation for the vector W (involving three grids in the vertical direction) is thus solved (Satoh 2002). This means that W is not advanced by linear additions in the acoustic

scheme. To insure the closure of the inline retrieval budget, we simply take the total changes that are contributed by the implicit solver in the acoustic scheme as acoustic modes of W in the third RK3 step. Note that this way does not violate the original W equation in Eq. (6). The contribution from these accumulated acoustic-step modes in the V and W tendency budgets are combined with their large-step PGF and PGFBUOY, respectively. Finally, we add the inline calculation for the tendency term outside of the RK3 integration loop, after the microphysics scheme:

$$\frac{\partial \Phi^{t+\Delta t}}{\partial t} \equiv \frac{\Phi^{t+\Delta t} - \Phi^t}{\Delta t} \tag{7}$$

where $\Delta t$ is the model integration time step and $\Phi$ represents V or W. The values of $\Phi$ at times $t$ and $t + \Delta t$, the latter denoted by superscripts, are termed the current and predicted states, respectively.

Figures 3 and 4 present the results of the inline budget analysis for horizontal momentum and vertical momentum, respectively, at three selected instants of 6, 12 and 16 hours. To demonstrate the momentum changes in a common physical

unit (velocities; $ms^{-1}$), every term of the flux-form budget equation shown in this paper is divided by the dry-air mass, $\mu_d{}^{t+\Delta t}$. The magnitude of the V tendency intensifies during this period with local maxima on the order of $10^{-4}$ to $10^{-3}$ $ms^{-2}$ (Fig. 3). Two forcing terms, PGF and COR, are a few times larger than the ADV term but generally offset each other, making the ADV term of comparable importance in determining the tendency. The CUV term for V tendency is generally small and thus not shown in Fig. (3). The residual, obtained from Eq. (1) with $P_V$ equal to zero, is always smaller than

$10^{-7}$ $ms^{-2}$ during the entire two-day simulation. The evolution of the 99th percentile residual term is shown in Fig. 5 (the





inline retrieval method is indicated by the black solid line), which reaches a value of about $3 \times 10^{-8}$ m$s^{-2}$ at around 15 hours. Meanwhile, the 99th percentile value of v tendency has a peak of $7 \times 10^{-4}$ m$s^{-2}$ (Fig. 2b). Thus, the relative magnitude of the 99th percentile residual is about 0.004% of the 99th percentile tendency term during the peak intensifying stage. Compared to the V tendency, the W tendency exhibits narrower features in the horizontal direction with an overall

smaller magnitude on the order of $10^{-9}$ m$s^{-2}$ (Fig. 4). The two largest forcings, PGFBUOY and CUV, usually have opposite signs, so their combined effect is on the same order as the ADV and the W tendency term. While the contribution from the upper-layer vertical velocity damping is not shown in Fig. 4 as it is generally small in the low layers, it is included as part of the rhs ($P_W$) when calculating the residual for the inline budget analysis. The residual in the inline W budget from Eq. (2) is generally four orders of magnitude smaller than its tendency term. The 99% percentile residual for W budget is

about 0.003% of the 99% percentile w tendency during the peak intensifying stage of the convection (not shown).

### 3.2 Post-processed momentum budget analyses

### 3.2.1 Key features and methodologies

In contrast to extracting terms directly from the model during its integration, most of the studies in which the momentum budget analysis is conducted use the model output variables after the completion of the integration. Note that since the sub-

output-time-step information are not available between successive outputs, only the large-step forcing terms can be estimated in these post-processed budget analyses. Generally, the neglect of the acoustic-modes is expected to have little impacts on the results as the high-frequency modes are often considered meteorologically insignificant. Most of the studies did not reveal the complete details about how their analysis was done, so we cannot presume their methodologies and the possible errors. However, a few simplifications commonly made in the post-processed budget analyses may introduce errors that

result in deviations from the simulated results and thus a significant residual. Below we revisit the relevant features of the WRF model that should be considered and discuss how they might affect the post-processed budget if they are ignored. Then, the results are shown for different post-processed budget analyses with different simplifications (Table 1). The aim herein is to identify these potential errors hidden in the budget calculation and how severely they affect the resulting interpretation.

(a) Diagnosed tendency

In a post-processed budget analysis, the tendency term of a given variable is approximated by the difference between the value of this variable at two successive output divided by the output interval. Thus, the accuracy is sensitive to the output interval. The value at the predicted state has a form of

$$\left. \frac{\partial \Phi^{t+\Delta t}}{\partial \text{t}} \right|_{diagnosed} \approx \frac{\Phi^{t+\Delta t} - \Phi^{t+\Delta t-\Delta t_{output}}}{\Delta t_{output}}. \tag{8}$$





If the output interval is longer than the model integration time step, the diagnosed tendency would deviate from the model prediction of the instantaneous tendency. To increase the accuracy, the output time interval $\Delta t_{output}$ needs to be similar to the integration time step $\Delta t$.

(b) Spatial discretization on the C staggered grid

For computational efficiency and accuracy, WRF utilizes a C-grid staggering system in which some variables are
arranged on grids that are shifted by a half-grid point from the others as illustrated in Fig. 6. This staggering system is pertinent to the numerical solution for spatial derivatives, e.g., the pressure gradient and the advection. For most of the spatial derivatives other than advection, the second-order centered operator for finite differencing is used in the WRF model. For example, the y-derivative of $\Phi$ is calculated using the discrete operator:

$$\frac{\partial \Phi}{\partial y}_{i,j,k} = \frac{1}{\Delta y}\left(\Phi_{i,j+\frac{1}{2},k} - \Phi_{i,j-\frac{1}{2},k}\right). \tag{9}$$

The index $(i,j,k)$ corresponds to a variable location with $(x,y,\eta) = (i\Delta x, j\Delta y, k\Delta \eta)$, where the $\Delta x, \Delta y$ and $\Delta \eta$ are the grid lengths in the two horizontal and vertical directions, respectively. The same strategy applies for the $x$- or the $\eta$- derivatives. When variables are staggered, the first rule of thumb is that the locations indicated as $(i,j,k)$, $(i,j+\frac{1}{2},k)$ and $(i,j-\frac{1}{2},k)$ shown in Eq. (9) should be on the same grid as the "targeted" prognostic variable. That is, for example, to calculate budget analysis for the V tendency, all the rhs forcing terms should be estimated on the V-grid. Taking the PGF for V tendency as an example, to calculate the term $\frac{\partial p}{\partial y}$ in Eq. (1), pressure values at locations with indices of $(i,j-\frac{1}{2},k)$ and $(i,j+\frac{1}{2},k)$ on the
V-grid are required according to Eq. (9) (red arrows in Fig. 6a). An interpolation to the target location might be necessary before applying the discrete operator, as in the case for deriving the term $\frac{\partial p}{\partial \eta}$ in Eq. (1). In this case, the pressures at locations with indices of $(i,j,k-\frac{1}{2})$ and $(i,j,k+\frac{1}{2})$ on the V-grid are calculated by taking a linear interpolation using the closest four available values from their original grids (Fig. 6b), e.g.,

$$p_{V-grid\left(i,j,k-\frac{1}{2}\right)} = \frac{\frac{1}{2}\left(p_{p-grid(i,j-1,k)} + p_{p-grid(i,j,k)}\right)\frac{\Delta \eta_{k-1}}{2}}{\frac{1}{2}(\Delta \eta_k + \Delta \eta_{k-1})} + \frac{\frac{1}{2}\left(p_{p-grid(i,j-1,k-1)} + p_{p-grid(i,j,k-1)}\right)\frac{\Delta \eta_k}{2}}{\frac{1}{2}(\Delta \eta_k + \Delta \eta_{k-1})} \tag{10}$$

in which different weights are applied because of the irregular (stretched) vertical grid-lengths.

If the C-grid staggering is not considered during the post-processing analysis, i.e., all the variables have been interpolated on the universal grids before budget calculation, in addition to the potential errors brought on by the interpolation method, Eq. (9) would essentially involve $\Phi_{i,j+1,k}$ and $\Phi_{i,j-1,k}$ over a larger grid interval of $2 \times \Delta y$ instead of $\Delta y$ with larger
truncation errors.

(c) Advection operators





For advection, higher-order operators for finite differencing are provided as the default setup for the WRF users. Taking the y component of the flux-form advection for V momentum in Eq. (3) as an example, with a fifth-order operator as selected in the present simulation, it is written as:

$$
\qquad -\frac{\partial(V\mathrm{v})}{\partial y}\Bigg|_{i,j,k} \approx -\frac{1}{\Delta y}\left(V_{i,j+\frac{1}{2},k}\,\mathrm{v}^{5th}_{i,j+\frac{1}{2},k} - V_{i,j-\frac{1}{2},k}\,\mathrm{v}^{5th}_{i,j-\frac{1}{2},k}\right) \qquad (11)
$$

where V and v are the mass coupled- and uncoupled-velocities, respectively.

$$
\mathrm{v}_{i,j-\frac{1}{2},k}{}^{5th} = \mathrm{v}_{i,j-\frac{1}{2},k}{}^{6th} - \mathrm{sign}\left(V_{i,j-\frac{1}{2},k}\right)\frac{1}{60}\left[\left(\mathrm{v}_{i,j+2,k}-\mathrm{v}_{i,j-3,k}\right)-5\left(\mathrm{v}_{i,j+1,k}-\mathrm{v}_{i,j-2,k}\right)+10\left(\mathrm{v}_{i,j,k}-\mathrm{v}_{i,j-1,k}\right)\right]
$$

and

$$
\mathrm{v}_{i,j-\frac{1}{2},k}{}^{6th} = \frac{1}{60}\left[37\left(\mathrm{v}_{i,j,k}-\mathrm{v}_{i,j-1,k}\right)-8\left(\mathrm{v}_{i,j+1,k}+\mathrm{v}_{i,j-2,k}\right)+1\left(\mathrm{v}_{i,j+2,k}+\mathrm{v}_{i,j-3,k}\right)\right]
$$


The odd-order advection operators include a spatially centered even-order operator and an upwind diffusion term. A detailed discussion on the advection scheme in the WRF model with different-order operators can be found in Wicker and Skamarock (2002) and Skamarock et al. (2008). Simplifying the advection estimation using an operator with order that differs from the numerical setup would indicate the neglect of some terms and a resulting source of errors in the ADV estimation.

(d) Forward/backward Euler method

Conceptually, the WRF model can be considered more of a forward scheme, i.e., using the known variables from the current state to calculate the forcing and then advancing the variables forward until reaching the predicting time. However, there are a few implicit components during the integration. For example, as discussed in Sect. 2.1, the large-step forcings are updated using a predictor-corrector method in the second and third RK3 steps. In addition, the W equation is coupled with
the geopotential tendency equation and includes a forward-in-time weighting that utilizes predicted states of the geopotential and temperature in solving the W (Eqs. (3.11), (3.12) and (3.19) in Skamarock et al., 2008).

In numerical analysis for solving ordinary differential equations, the (explicit) forward Euler method approximates the change of a system from $t$ to $t + \Delta t$ using the current states ($t$) while the (implicit) backward Euler method finds the solution using the predicted states ($t + \Delta t$):

$$
\qquad \frac{\partial \Phi^{t+\Delta t}}{\partial t} \approx F(\Phi^{t}) \quad \text{forward Euler method} \qquad (12)
$$

$$
\frac{\partial \Phi^{t+\Delta t}}{\partial t} \approx F(\Phi^{t+\Delta t}) \quad \text{backward Euler method} \qquad (13)
$$

Consistent with this concept, the rhs forcing terms of a budget equation can be estimated using two different instantaneous states in analogous ways. (However, we emphasize that the post-processed budget analysis does not solve the tendency equation per se but only diagnoses the relationship between the two sides of the equation.) Note that for post-
processing analyses, the availability of the data depends on the output time interval ($\Delta t_{output}$), which is often much larger than the integration time step ($\Delta t$). Thus, for the tendency at a given time $t + \Delta t$, when applying the forward Euler method





to estimate the associated rhs forcings, the "current states" one can use are the most recent prior output at $t + \Delta t - \Delta t_{output}$ (see Fig. 7):

$$\frac{\partial \Phi^{t+\Delta t}}{\partial t} \approx F\big(\Phi^{t+\Delta t-\Delta t_{output}}\big) \quad \text{forward Euler method for post} - \text{processing} \tag{14}$$

If $\Delta t_{output}$ is the same as $\Delta t$, Eq. (14) reverts to Eq. (12). If $\Delta t_{output}$ is much larger than $\Delta t$, the backward Euler method using predicted states at $t + \Delta t$ may better estimate the true model forcing terms as they are calculated using variables at a closer time to the real integration window in the model (Fig. 7).

The above two diagnostic methods estimate the forcing terms using instantaneous states. However, as mentioned in Sect. 3.2.1(a), the diagnosed lhs tendency depends on two successive output. Thus, an average between forcings diagnosed
explicitly and implicitly are often considered. For a post-processed analysis, this translates into estimating the forcings using both predicted states and the most recent prior available current states:

$$\frac{\partial \Phi^{t+\Delta t}}{\partial t}\bigg|_{diagnosed} \approx \frac{1}{2}[F\big(\Phi^{t+\Delta t-\Delta t_{output}}\big) + F(\Phi^{t+\Delta t})]. \tag{15}$$

(e) Flux or advective form of equation

While the momentum equations solved in the WRF model are in the flux form, their corresponding advective forms can
be derived and are often used for post-processed budget analyses for convenience. To derive the advective form, the flux-form V momentum equation (Eq. (1) excluding residual) is first multiplied by a factor of $\frac{1}{\mu_d}$ and V is rewritten as $\mu_d v$:

$$\underbrace{\frac{1}{\mu_d}\frac{\partial(\mu_d v)}{\partial t}}_{\substack{\text{V tendency}}} = \underbrace{-\frac{1}{\mu_d}\nabla \cdot (\mu_d \vec{v}v)}_{\substack{\text{advection} \\ \text{ADV}}} + \underbrace{\frac{1}{\mu_d}\left[-\alpha\frac{\partial p}{\partial y} - \frac{\alpha}{\alpha_d}\frac{\partial p}{\partial \eta}\frac{\partial \phi}{\partial y}\right]}_{\substack{\text{horizontal pressure gradient force} \\ \text{PGF}}} \underbrace{-\frac{1}{\mu_d}fU}_{\substack{\text{Coriolis} \\ \text{COR}}} \underbrace{-\frac{1}{\mu_d}\left(\frac{vW}{r_e}\right)}_{\substack{\text{curvature} \\ \text{CUV}}} + \underbrace{\frac{1}{\mu_d}P_V}_{\substack{\text{remaining} \\ \text{(parameterized)} \\ \text{physics}}} \tag{16}$$

Then, by adding the mass continuity equation in WRF (multiplied by a factor of $\frac{v}{\mu_d}$):

$$\frac{v}{\mu_d}\left[\frac{\partial \mu_d}{\partial t} + \nabla \cdot (\mu_d \vec{v})\right] = 0$$

to the RHS of Eq. (16), we obtain

$$\underbrace{\frac{1}{\mu_d}\frac{\partial(\mu_d v)}{\partial t}}_{\substack{\text{V tendency}}} = \frac{v}{\mu_d}\frac{\partial \mu_d}{\partial t} + \underbrace{\frac{v}{\mu_d}\nabla \cdot (\mu_d \vec{v}) - \frac{1}{\mu_d}\nabla \cdot (\mu_d \vec{v}v)}_{\substack{\text{advection} \\ \text{ADV}}} + \underbrace{\frac{1}{\mu_d}\left[-\alpha\frac{\partial p}{\partial y} - \frac{\alpha}{\alpha_d}\frac{\partial p}{\partial \eta}\frac{\partial \phi}{\partial y}\right]}_{\substack{\text{horizontal pressure gradient force} \\ \text{PGF}}} \underbrace{-\frac{1}{\mu_d}fU}_{\substack{\text{Coriolis} \\ \text{COR}}} \underbrace{-\frac{1}{\mu_d}\left(\frac{vW}{r_e}\right)}_{\substack{\text{curvature} \\ \text{CUV}}} + \underbrace{\frac{1}{\mu_d}P_V}_{\substack{\text{remaining} \\ \text{(parameterized)} \\ \text{physics}}} \tag{17}$$

Moving the first term on the rhs of Eq. (17) is to the lhs, the second rhs term can be combined with flux-form advection using the vector identity $\nabla \cdot (\mu_d \vec{v}) = \mu_d(\nabla \cdot \vec{v}) + \vec{v} \cdot (\nabla \mu_d)$. Then, the advective form of the horizontal momentum equation
is obtained as:





$$\underbrace{\frac{\partial v}{\partial t}}_{\substack{\text{v tendency} \\ \text{in advective form}}} = \underbrace{-\vec{v} \cdot \nabla v}_{\substack{\text{advection ADV} \\ \text{in advective form}}} + \underbrace{\frac{1}{\mu_d}\left[-\alpha \frac{\partial p}{\partial y} - \frac{\alpha}{\alpha_d}\frac{\partial p}{\partial \eta}\frac{\partial \phi}{\partial y}\right]}_{\substack{\text{horizontal pressure gradient force} \\ \text{PGF}}} - \underbrace{\frac{1}{\mu_d}fU}_{\substack{\text{Coriolis} \\ \text{COR}}} - \underbrace{\frac{1}{\mu_d}\left(\frac{vW}{r_e}\right)}_{\substack{\text{curvature} \\ \text{CUV}}} + \underbrace{\frac{1}{\mu_d}P_V}_{\substack{\text{remaining} \\ \text{(parameterized)} \\ \text{physics}}} \qquad (18)$$

### 3.2.2 Results- horizontal momentum budget

Table 1 summarizes all the post-processed budget analyses tested in this study. In the present section, we first present the
results one by one, and a qualitative inter-comparison among them and the inline retrieval method is discussed. The first
post-processed method (POST10min-E) for V budget follows all the approaches in the model as closely as possible using the
10-min output data. The flux-form equation, C staggering grids, and the same orders of advection operators as the
experimental setup are used. The diagnosis of the large-step forcing is applied directly on the model outputs on $\eta$ levels
using the explicit/forward Euler method as shown in Eq. (14). The diagnosed forcing terms are compared with their
corresponding true values from the inline retrieval (Fig. 8). Errors smaller than, but on the same order of $10^{-4}\ m\ s^{-2}$ as V
tendency, are observed in all terms including the diagnosed tendency term. These errors grow in magnitude and areal
coverage with the growth of the disturbance. Aside from COR, the absolute errors in the tendency, ADV and PGF can
exceed $6 \times 10^{-4}\ m\ s^{-2}$, the former two of which are more than 50% of the magnitude of their true values locally.

The second post-processed analysis (POST1min-E) is done following the same approach but applied on the 1-min (same
as the integration time step for this simulation) output data, and the results show strongly reduced errors in all terms (Fig. 9).
The errors that remain are mostly in the PGF term and likely stem from the fact that the acoustic-step modes and the RK3
integration scheme are not considered in the post-processed budget. These inherent errors result in a small residual term with
a general order of $10^{-5}\ m\ s^{-2}$, one to two order(s) smaller than the maximum v tendency. However, in terms of local
maxima, the 99th percentile residual obtained in POST1min-E can reach a relative magnitude of about 30% of the 99th
percentile v tendency during the peak intensifying stage of the convection at around 15 hours (Figs. 2b and 5). Although
reducing the model output interval to be close to the integration time step helps to balance the budget without the need for
inline diagnoses, it is computationally expensive especially for large, data-intensive simulations.

Given that computational cost is often a major consideration, we also test whether the implicit/backward Euler method
(POST10min-I) can improve the estimation of instantaneous forcing terms relative to the explicit method for the same 10-
min output data (POST10min-E). POST10min-I follows the same strategy as POST10min-E except that all the rhs terms,
following Eq. (13), are diagnosed with the predicted states instead of the previous output states. As depicted in Fig. 10,
POST10min-I indeed better captures the true model estimated forcing values as errors in all the forcing terms diminish
greatly to an accuracy similar to POST1min-E. However, as these forcings are calculated at a given instant, the imbalance of
the budget remains as the diagnosed tendency term is not calculated instantaneously. If budget analysis at an instant of time
is desired, we recommend adding the tendency calculation within the model as a standard output and diagnosing the forcing





terms implicitly, which yields a residual term on the similar order to the one obtained in POST1min-E (the rightmost column in Fig. 10; Fig. 5).

For the more common situation where the post-processed analyses diagnose tendency terms using two outputs over an output time interval, the averaged forcings calculated explicitly and implicitly are tested using Eq. (15) on the 10-min output

(POST10min-(E+I)/2). The residual obtained from this method shows a similar accuracy level as POST1min-E and POST10min-I (Figs. 11 and 5).

We now investigate the impact of other common simplifications on top of the reference experiment, POST10min-(E+I)/2. The first such simplification is to approximate the flux-form advection term using the second order operator (Eq. (9)) for both vertical and horizontal components (POST2oadv-(E+I)/2) instead of the third and fifth order operators as used

in the model setup. In our simulation, such inconsistency of advection operators introduced errors in the ADV term with a maximum value $>3 \times 10^{-4} \ m \ s^{-2}$, more than 50% of its true magnitude along the slantwise convective band (Fig. 12). Next, we repeated POST10min-(E+I)/2 but applied to the output data that have been interpolated to the universal/un-staggered grid same as the one for the pressure variable (p-grid) (POSTnonstag-(E+I)/2). This is a common way to post-process model output data for plotting purposes. As mentioned earlier, this approach would reduce the accuracy when solving the spatial

differential terms and indeed the results indicate significant errors over a large area in both ADV and PGF (Fig. 13). Their combined errors result in widespread residual values $>3 \times 10^{-5} \ m \ s^{-2}$ even over the area where the tendency term is smaller than $1 \times 10^{-4} \ m \ s^{-2}$ (error is at least of 30% magnitude of the tendency term locally).

Finally, a totally different format of V equation, the advective form, is used for post-processed analysis (POSTadvF-(E+I)/2). Mathematically, the flux-from momentum equation can be rewritten in the advective form without making any

additional approximation, only with the aid of the conservation law of dry-air mass in the WRF model as shown in Eqs. (16)-(18). However, during the interchange of the expression for the tendency and advection terms, truncation errors may be introduced. We reiterate that the tendency term in the advective form is not equivalent to the one in the flux form divided by $\mu_d$, however, calculation suggests that they are approximately equal

$$\frac{1}{\mu_d}\frac{\partial(\mu_d \mathrm{v})}{\partial \mathrm{t}} \approx \frac{\partial \mathrm{v}}{\partial \mathrm{t}}$$

with a maximum error that is on the order of $10^{-7} \sim 10^{-8} \ m \ s^{-2}$ (three orders of magnitude smaller than the v tendency) in our study. The summation of the tendency term and advection term in these two forms of the momentum equation should be mathematically identical, so we would expect to see a small difference in the advection term as in the tendency term. However, we find that the advection term in the advective-form has a strong positive bias as compared to the one in the flux form (Fig.14). The residual term in the POSTadvF-(E+I)/2 is thus negatively biased over the entire convective band with a

magnitude exceeding $1.2 \times 10^{-4} \ m \ s^{-2}$ (reaching 100% error near the upper half of the convective band).

A quantitative comparison of the magnitude of the 99% residual term in the domain (excluding the boundaries) among different analysis methods is shown in Fig. 5. The residuals between the instantaneously-diagnosed forcings and the true model tendency term (calculated inline) are shown in Fig. 5a while the ones between the averaged forcings of two





consecutive outputs and the diagnosed tendency term are shown in Fig. 5b. The evolution of the 99th percentile residual

shows generally larger values when the momentum tendency is larger (Fig. 2b), suggesting that these errors may amplify in stronger convection cases. While the post-processed budget analysis in POST1min-E, POST10min-I and POST10min-(E+I)/2 can achieve a relatively small 99th percentile residual ($\sim 2 \times 10^{-4}\ m\ s^{-2}$ , or about <30% of the tendency, at the peak intensifying stage), the inline budget analysis always gives a much smaller value ($< 10^{-7}\ m\ s^{-2}$, or <0.004% of the tendency, during the entire simulation). Any other simplification can severely impair the accuracy of the post-processed

budget analysis. Both POSTnonstag-(E+I)/2 and POSTadvF-(E+I)/2 can lead to a 99th percentile residual term larger than $10^{-3}\ m\ s^{-2}$, peaking at around $2.2 \times 10^{-3}\ m\ s^{-2}$ and $5 \times 10^{-3}\ m\ s^{-2}$, which correspond to >300% and >800% of their concurrent 99th percentile v tendency, respectively.

### 3.2.3 Results- vertical momentum budget

For the W equation, the closure of the post-processed budget appears not practicable even when the output time interval

is reduced to the integration time step. This is likely because a large portion of model simulated PGFBUOY comes from the implicitly-solved acoustic modes which include vertically propagating sound waves and buoyancy oscillations, as shown in our inline budget retrieval (Fig. 4). These acoustic modes are spatially noisy over the growing band with a general order of $10^{-4}\ ms^{-2}$, which is one order of magnitude larger than the W tendency (see the blue and red contours overlapped on the residual subplots in Fig. 4). The application of POST1min-E for the W tendency shows that this method accurately estimates

most of the processes, but large errors $>2 \times 10^{-3}\ ms^{-2}$ remain in the PGFBUOY term resulting in a widespread residual that reaches the same magnitude of the peak W tendency term (Fig.15). The fact that these errors exceed the acoustic component of PGFBUOY suggests that such imbalance does not solely come from the neglect of the acoustic modes. A close comparison of the post-processed and the inline PGFBUOY shows that our estimation is close to the inline value to an accuracy of at least three significant figures at the first RK3 step before the acoustic contribution is considered (not shown).

However, this large-step forcing term adjusts rapidly, sometime even with a sign change, from step to step within the RK3 integration. Although it is feasible to estimate $F(\Phi^t)$ via post-processing, it is however impossible to retrieve $F(\Phi^{**})$ in Eq. (4), leading to the poor estimation of vertical pressure gradient and buoyancy force in the W budget.

### 4 Tests on different cases or with different horizontal resolutions

The growth of the residual as the convection intensifies (Fig. 5) motivates a test on a different case with stronger

momentum tendencies. A WRF idealized 2-D squall-line test case (em_squall2d_y; Skamarock et al., 2008) is selected with a horizontal resolution of 250 m and 3-second integration time step, and the simulation is integrated for 1 hour. A diffusion scheme based on a prognostic turbulent kinetic energy equation is activated (Skamarock et al., 2008, chapter 4.2.4). The simulated v tendency in this case is two orders of magnitude stronger than the one in the slantwise convection case. The inline retrieval budget tool works well with 99th percentile residuals generally two orders smaller than the tendency terms in

the domain (and a locally peak residual of <0.005 % of the peak tendency term) during this simulation. However, the post-



processed budget analysis applied on the output data with an output interval the same as the integration time step (analogous to POST1min-E but in this case, it is termed POST3sec-E; Fig. 16), with no simplification made, does not work as well as in the slantwise convection case. While the activated diffusion process is usually difficult to estimate via post-processing and so would be expected to result in some errors, the impact of subgrid turbulence is relatively small at such fine grid spacing (e.g.,

Bryan et al., 2003). Moreover, POST3sec-E shows that the largest error appears in the PGF term with a magnitude of 50% of its true value at a given instant. The error in diffusion only accounts for about 10% of the error at the same time. One possible reason is that unlike the case of slantwise convection where the PGF exhibits rather horizontally uniform structure with almost the same sign (Fig. 3), the PGF term in this case has a more complex spatial structure with several sign changes over a horizontal distance of 10-15 km. Thus, large errors appear at the edge of these positive/negative patches where the

sign changes. Despite the small spatial scales of these errors, the large error magnitude would render accurate interpretation of the physical process difficult based on such post-processed budget analysis. This result suggests that the post-processed budgets, even when done with care, do not always work well, and that the associated residual/errors might be sensitive to the intensity of the simulated system, the spatial/temporal resolution, and the nature of the physical processes governing the different systems.

While an increase in spatial resolution often requires a shorter integration time step for numerical stability and may result in stronger simulated convection, it is almost impossible to separate all these factors. We can, however, conduct the same slantwise convection simulation with a higher resolution of 2 km (and a shorter integration time step of 10 seconds) to exclude the effect of different physical processes and discuss the changes in the accuracy of the budget analysis when spatial resolution is increased from 10 km. As shown in Fig. 2b, in the 2-km simulation the maximum of the simulated $99_{th}$

percentile v tendency is $1.2 \times 10^{-3}$ m$s^{-2}$, almost twice the value in the 10-km run. The magnitude of the residual from the inline budget analysis also becomes larger with the $99_{th}$ percentile value almost one order larger than that in the 10-km simulation (Fig. 5). However, its relative magnitude is still small and amounts to about 0.02% of the tendency in the 2-km case. For the post-processed budget analysis applied on the 2-km simulation, the instantaneous calculation in POST10min-I-2km indicates large fluctuation in the $99_{th}$ percentile residual with time (Fig. 5a). While the low values remain similarly

small as those in its 10-km case, the peak values can be two to six times larger. For the method using two model outputs for both diagnosed tendency and forcing terms, the peak $99_{th}$ percentile residual in POST10min-(E+I)/2-2km is about ten times larger than that in its 10-km counterpart (POST10min-(E+I)/2). This is likely due to the larger deviation caused by the longer diagnosed window (10 minutes) with respect to the integration time step (10 seconds) in the 2-km case. In addition, it appears that the simulated fields adjust more rapidly with more complex structures on smaller scales in the 2-km simulation

as compared to the 10-km simulation (not shown). If the same analysis is performed using the 1-min output (POST1min-(E+I)/2-2km) as opposed to the 10-min output, the residual can be greatly reduced to only two time larger than the residual in POST10min-(E+I)/2 (Fig. 5).

The results presented above suggest that the absolute errors in both the inline and post-processed momentum budget analyses indeed increase with increasing horizontal resolution, although the relative magnitude with respect to the simulated





tendency does not increase substantially for the inline budget analysis. The accuracy of the post-processed budget analysis using the averages of two consecutive model outputs is highly dependent on the ratio of the output interval and the integration time step. Our results suggest that this ratio should be at most 10 (10-km case) and should be even smaller for high-resolution simulations (e.g., 6 for the 2-km case). For cases with a more complex physical process like the squall line test case, the inline budget retrieval appears necessary for adequate budget closure.

**5 Discussion and summary**

      Budget analysis is a commonly-employed tool in numerical studies to understand the underlying mechanisms for certain simulated features of interests. However, to the authors' knowledge many studies still have difficulties in achieving a balanced/closed budget especially when a full physics model is used and when the budget is calculated instantaneously over a local area. Aside from the complexity of various (some implicit) parameterization schemes, the main challenge in closing

the budget in a full-physics involves the analysis of post-processed data using algorithms that are inconsistent with the model solver. In this study, an inline momentum budget retrieval tool is developed for the WRF model, and its advantages for the momentum budget analysis are demonstrated. The 99th percentile residual obtained from this inline retrieval is always smaller than 0.02% of the tendency term in all the tested cases, which include idealized, 2D simulations of slantwise convection and squall lines. Taking the results from the inline retrieval as "truth", we investigate the potential errors in each

term and the resultant residual for post-processed budget analyses under different assumptions.

      The comparison among different post-processed diagnoses is focused on the horizontal momentum (V) budget. The reason is that post-processed vertical momentum (W) budget analysis fails to produce reasonably accurate results due to the noisy vertical pressure gradient and buoyancy forces that are tied closely to the acoustic modes and the implicit scheme used for the vertical momentum integration. Thus, inline retrieval is necessary for an accurate W budget analysis. The errors in the

post-processed V budget diagnose arise from both the left-hand-side tendency term and the right-hand-side (rhs) forcing terms. To improve the accuracy of the diagnosed momentum tendency estimation, one can reduce the output interval to the model integration time step, which incurs a large computational cost. An alternative and cheaper solution is to add the tendency calculation within the model as a standard output. Our test case of slantwise convection shows that the diagnosed tendency using two successive model outputs with a 10-min interval to approximate the instantaneous true tendency (with an

integration time step of 1 minute) could create an error exceeding 50%.

      For the rhs forcing terms in the V equation, errors can be limited if the post-diagnosis is done with care using the same form of the model equation, the same spatial discretization, the same order of the advection operators, and performing the calculation on the original (e.g., C-staggering and vertically stretched) model grids. However, the above steps are necessary but may not be sufficient for the closure of the budget as the forcing term diagnosis also largely depends on the selected

input states. If the budget at an instant of time is desired, the explicit/forward Euler method using the previous states might result in large and widespread errors in the advection and horizontal pressure gradient terms (local peak errors are about 50% and 25% of their true values in our simulation, POST10min-E) unless the output interval is reduced to the integration time

step. In the latter case (POST1min-E), an error < 5% for each individual term and a residual generally one to two order(s) smaller than the maximum tendency can be achieved (although locally the 99th percentile residual reaches to almost 30% of

the 99th percentile v tendency). An alternative way to reach a similar level of accuracy without compromising the computational cost is to diagnose the rhs forcings using the implicit/backward Euler method (POST1min-I). This method diagnoses the instantaneous forcings using the predicted states and thus can better capture the true model forcings by using inputs at a closer time to the model integration window.

Instead of performing the calculation using model output at one given instant, a more general post-processed budget

analysis can use two successive model outputs (POST10min-(E+I)/2). This method seems to work well with a residual term generally one to two order(s) smaller than the maximum tendency in our 10-km case with 10-min output intervals. However, the accuracy of this method varies among the test cases and is sensitive to the ratio of the output interval to the integration time step. Among the tests conducted in this study, an upper limit of 10 for this ratio is suggested, and it should be even smaller for high-resolution simulations of high-amplitude weather systems, as rapid adjustments occur on the small scales.

Three other common assumptions in post-processing analysis are made on top of the POST10min-(E+I)/2 to examine their potential impacts on the accuracy of the horizontal momentum budget analysis. First, utilizing an advection operator with a lower order than the one used in the model setup impairs the accuracy of the advection term with up to 50% error over the area where the advection is the strongest (POST2oadv-(E+I)/2). Second, the neglect of the staggering grids would negatively impact the estimation of all the spatial differential terms, leading to a widespread residual of at least 30% of the

local tendency (POSTnonstag-I). Its maximum 99th percentile residual is even three times larger than the 99th percentile momentum tendency term at the strongest intensifying stage. Last, when the advective form of the momentum equation is used for post-diagnosis rather than the flux form, although it is mathematically equivalent to the flux form solved in the model, a negatively biased residual results with a peak 99th percentile value almost eight times of the tendency term (POSTadvF-I). All the above errors do not just appear randomly; rather, they are spread over the area where the dynamics

are the most active, thus undermining the physical interpretation on the dynamics of the simulated system. We thus emphasize the importance of revealing the magnitude of the residual (relative to the tendency term) in publications on budget analysis, to enable readers to gauge the validity of the results.

While the post-processed V budget analysis can reach an acceptable accuracy in some cases, the residual increases with increasing spatial resolution and the results vary from case to case even when the same analysis method is adopted. Our test

of an idealized squall line case with strong V tendencies shows that the application of the post-processed budget analysis method without any simplification using the 3-second output data nevertheless results in large magnitude of errors (~50%) in the horizontal pressure gradient force, with very small-scale error structures.

In summary, this study has shown how different assumptions/simplifications made in a post-processed budget analysis may severely impact the estimation of each forcing term and result in a large imbalance of the budget. Based on our

experiments, we conclude that the inline retrieval method like that developed herein is the most reliable one for budget analysis in numerical studies. While the budget analyses shown in this study are only for V and W momentum under the 2D



idealized configurations, this developed tool also retrieves budget terms for U momentum and potential temperature and can be applied to 3D idealized and real cases. We also stress that in some budget studies where a coordinate transformation is necessary (e.g. from Cartesian to polar), some errors are unavoidable. In such cases, it is best to perform the budget

calculation using the inline retrieval method on the model grid and then transform the budget to a new coordinate (e.g., Zhang et al., 2000). Finally, in situations where the inline coding cannot be done, this study also provides general guidance to minimize the error in the budget. Thus, our results are beneficial to budget analyses in numerical studies in general, and not limited to the WRF model.


**Code availability**

The standard version of WRF v3.8.1 is publicly available at http://www2.mmm.ucar.edu/wrf/users/download/get_sources.html. The inline budget retrieval tool in the WRF v3.8.1 described in this study can be found at https://github.com/ting-chenCHEN/WRFV3.8.1_inline_budget_retrieval (the version

for this study is tagged GMD_submission1) or https://zenodo.org/record/3373872.

**Appendix**

To construct an initial condition that contains conditional symmetric instability (CSI) but to avoid dry symmetric instability and dry and conditional (gravitational) instability is a challenging task as discussed in Persson and Warner (1995). Therefore, the initial profile in our test case is decided by a trail-and-error method and follows the following steps:

(1) We first prescribe a horizontally uniform Brunt-Vaisaila frequency, $N^2 = \frac{g}{\theta_v}\frac{\partial \theta_v}{\partial z}$ with a vertical profile of

$$N^2 = \begin{cases} 1.25 \times 10^{-4} & z < 0.5 \ km \\ 9 \times 10^{-5} & s^{-2} \ \text{for} \ 5km \ \leq z < 10.5 \ km \\ 5 \times 10^{-4} & z \geq 13.5 \ km \end{cases} \quad (A1)$$

       where z is the height and there is a linear transition for layers where $0.5 \ km \leq z < 5 \ km$ and $10.5m \leq z < 13.5 \ km$ using the specified values beneath to above them.

    (2) A constant geostrophic vertical zonal wind shear is given, $\frac{\partial U_g}{\partial z} = 5.8 \times 10^{-3} \ s^{-1}$. The thermal wind balance gives

$$\frac{\partial U_g}{\partial z} = -\frac{g}{f\theta_v}\frac{\partial \theta_v}{\partial y} \quad (A2)$$

    (3) Base on (A1) and (A2), we can specify the value of $\theta_v$ at any one point and then derive the $\theta_v$ for the entire domain. In this case, $\theta_v(y_0, z_0) = 287.5K$ where $(y_0, z_0)$ indicates the grid at the surface on the southernmost boundary.

    (4) The relative humidity (RH) field is constructed by specifying a horizontally uniform background profile

540        ($RH_{background}$) with some enhancement ($RH_{bubble}$) over an elliptical area where the initial perturbation will be


later inserted. The reasons for the enhanced humidity over a limited area are to hasten the release of CSI and to avoid convection from developing near the southern boundary.

$$RH_{background}(z) = \begin{cases} 0.81 & z \leq 5km \\ \min\left[0.81, 1 - 0.9\left(\frac{z-5}{7.5}\right)^{0.8}\right] & \text{for } 5km < z < 12.5\ km \\ 0.1 & z \geq 12.5\ km \end{cases}$$

$$RH_{bubble}(y, z) = RH_{background}(z) \times f_{enhancement}(y, z),$$

where

$$f_{enhancement}(y, z) = \begin{cases} 1.22 & e \leq 1 \\ 1.22 - 0.11(e-1) & \text{for } 1 < e \leq 3, \\ 1 & e > 3 \end{cases}$$

where $e = (\frac{y-410}{e_b})^2 + (\frac{z-1}{e_a})^2$, $e_b = 100$, $e_a = 3$ and y and z are the horizontal distance from the southern boundary and height, respectively, with units of km. The constructed initial profile has a maximum RH of 98.82% over an elliptical area centered at y = 410 km and z =1 km.

(5)  A constant surface pressure is specified, $P_{sfc} = 1000\ hPa$.

(6)  We then iteratively solve for the hydrostatically balanced pressure, water vapor mixing ratio, potential temperature, dry and full (moist) air density, and geostrophic zonal wind for the entire domain.

The constructed initial environment contains some CSI, which is identified by the presence of negative saturated geostrophic potential vorticity (Chen et al. 2018). In this test case, CSI only exists over the southern-half of the domain and never extends to higher than 5 km.


To initiate convection, a 2-D bubble of maximum potential temperature perturbation $\Delta\theta = 0.5$ K and zonal wind perturbation $\Delta u = -6\ ms^{-1}$ is inserted in the area where RH is maximized and where the saturated geostrophic potential vorticity has a value of about -0.2~-0.1 pvu. This perturbation bubble has a maximum width of 100 km and thickness of 3 km, with its center located at y = 400 km and z =1.5 km.


**Author contribution**

TC designed and performed the numerical experiments under the supervision of MY and DK. MY proposed the idea of comparing the inline and post-processing budget analyses. TC developed the code of the inline budget retrieval tool in the WRF v3.8.1 model and the post-processing analyses. DK provided useful suggestions to improve the work. TC prepared the manuscript and all co-authors contributed to the writing and editing of the paper.


**Competing interests**

The authors declare that they have no conflict of interest.

**Acknowledgements**





The research reported here has been supported by the Fonds de recherche du Québec – Nature et technologies (FRQNT) doctoral research scholarship and the NSERC/Hydro-Quebec Industrial Research Chair program.

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





670

**Table 1. A summary of all different approaches for the post-processed horizontal momentum budget analysis that are applied on the model output after the integration finishes.**

| | Form of the equation | Output time interval | Order of (vertical; horizontal) advection operators | Forcing terms diagnosed using the explicit or implicit method | Calculated on C staggering grids |
|---|---|---|---|---|---|
| Slantwise convection simulation with a grid length of 10 km and integration time step of 1min. | | | | | |
| POST10min-E | Flux form | 10 mins | 3; 5 | Explicit | Yes |
| POST1min-E | Flux form | 1 min | 3; 5 | Explicit | Yes |
| POST10min-I | Flux form | 10 mins | 3; 5 | Implicit | Yes |
| POST10min-(E+I)/2 | Flux form | 10 mins | 3; 5 | Average of explicit and implicit | Yes |
| POST2oadv-(E+I)/2 | Flux form | 10 mins | 2; 2 | Average of explicit and implicit | Yes |
| POSTnonstag-(E+I)/2 | Flux form | 10 mins | 3; 5 | Average of explicit and implicit | No |
| POSTadvF-(E+I)/2 | Advective form | 10 mins | 3; 5 | Average of explicit and implicit | Yes |
| Slantwise convection simulation with a grid length of 2 km and integration time step of 10 secs. | | | | | |
| POST10min-I-2km | Flux form | 10 mins | 3; 5 | Implicit | Yes |
| POST10min-(E+I)/2-2km | Flux form | 10 mins | 3; 5 | Average of explicit and implicit | Yes |
| POST1min-(E+I)/2-2km | Flux form | 1 mins | 3; 5 | Average of explicit and implicit | Yes |
| Squall line simulation with a grid length of 250 m and integration time step of 3 secs. | | | | | |
| POST3sec-E | Flux form | 3 secs | 3; 5 | Explicit | Yes |





## WRF Model Integration from $\Phi^t$ (known) to $\Phi^{t+\Delta t}$

**RK3 time integration scheme:** *including advection, Coriolis, mixing... and all the parameterized physics

**(1) 1st RK3 step** ➡ **Calculate and store all* the large-step forcing terms (RHS), $F(\Phi^t)$**

➡ **Acoustic integration (one step, with $\Delta\tau = \Delta t/3$ )**

(A) define perturbation variables : for 1st RK3, $\Phi^\tau = \Phi^t$, $\Phi''^\tau = 0$
(B) then, $\Phi''^\tau$ is being advanced through

(a) 1st acoustic step

$$\Phi''^{\tau+\Delta\tau} = \Phi''^\tau + F(\Phi^t)\Delta t/3 + [ACOUS(\Phi''^\tau)]$$

➡ Recover perturbation field to full field

$$\Phi^{\tau+\Delta\tau} = \Phi''^{\tau+\Delta\tau} + \Phi^t = F(\Phi^t)\Delta t/3 + [ACOUS(\Phi''^\tau)] + \Phi^t \equiv \Phi^*$$

$\Phi^t$ —————→ $\Phi^*$ ············· $\Phi^{t+\Delta t}$

**(2) 2nd RK3 step** ➡ **UPDATE and store all the large-step forcing terms (RHS), $F(\Phi^*)$**

➡ **Acoustic integration (2 steps, with $\Delta\tau = \Delta t/4$)**

(A) define perturbation variables : $\Phi^\tau = \Phi^*$, $\Phi''^\tau = \Phi^t - \Phi^*$
(B) then, $\Phi''^\tau$ is being advanced through

(a) 1st acoustic step

$$\Phi''^{\tau+\Delta\tau} = \Phi''^\tau + F(\Phi^*)\Delta t/4 + [ACOUS(\Phi''^\tau)]$$

(b) 2nd acoustic step

$$\Phi''^{\tau+2\Delta\tau} = \Phi''^{\tau+\Delta\tau} + F(\Phi^*)\Delta t/4 + [ACOUS(\Phi''^{\tau+\Delta\tau})]$$

➡ Recover perturbation field to full field

$$\Phi^{\tau+2\Delta\tau} = \Phi''^{\tau+2\Delta\tau} + \Phi^*$$
$$= \Phi''^\tau + F(\Phi^*)\Delta t/2 + [ACOUS\ sum] + \Phi^*$$
$$= \Phi^t + F(\Phi^*)\Delta t/2 + [ACOUS\ sum] \equiv \Phi^{**}$$

$\Phi^t$ —→ $\Phi^*$ —→ $\Phi^{**}$ ············· $\Phi^{t+\Delta t}$

**(3) 3rd RK3 step** ➡ **UPDATE and store all the large-step forcing terms (RHS), $F(\Phi^{**})$**

➡ **Acoustic integration (4 steps, with $\Delta\tau = \Delta t/4$)**

(A), (B)...

(a)-(d) 1st -4th acoustic step ...

➡ Recover perturbation field to full field

$$\Phi^{t+\Delta t} = \Phi^t + F(\Phi^{**})\Delta t + [ACOUS\ sum]$$

$\Phi^t$ —→ $\Phi^*$ —→ $\Phi^{**}$ ——————→ $\Phi^{t+\Delta t}$

**Compute the contribution from microphysics physics and adjust $\Phi$**

675

**Figure 1: The time integration strategy for advancing a state variable (generalized as Φ) in the WRF model. In this given example, four acoustic steps are specified for one integration time. Adapted from Skamarock et al. (2008).**



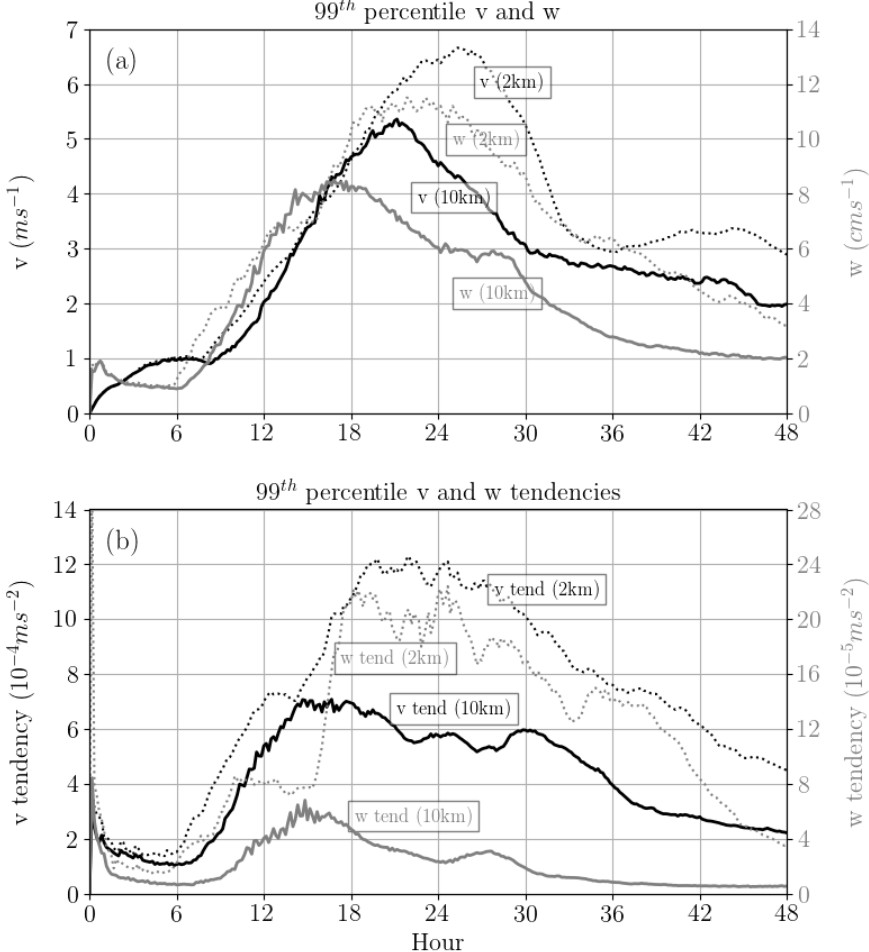

**Figure 2. Evolutions of the 99 percentiles of (a) horizontal velocity, v (black; axis on the left), and vertical velocity, w (gray; axis on the right) in the simulation of slantwise convection. (b) is same as (a) but for their tendencies (black and gray lines for v and w tendencies, respectively). Solid lines are for the 10-km simulation while the dotted ones are for the 2-km case.**

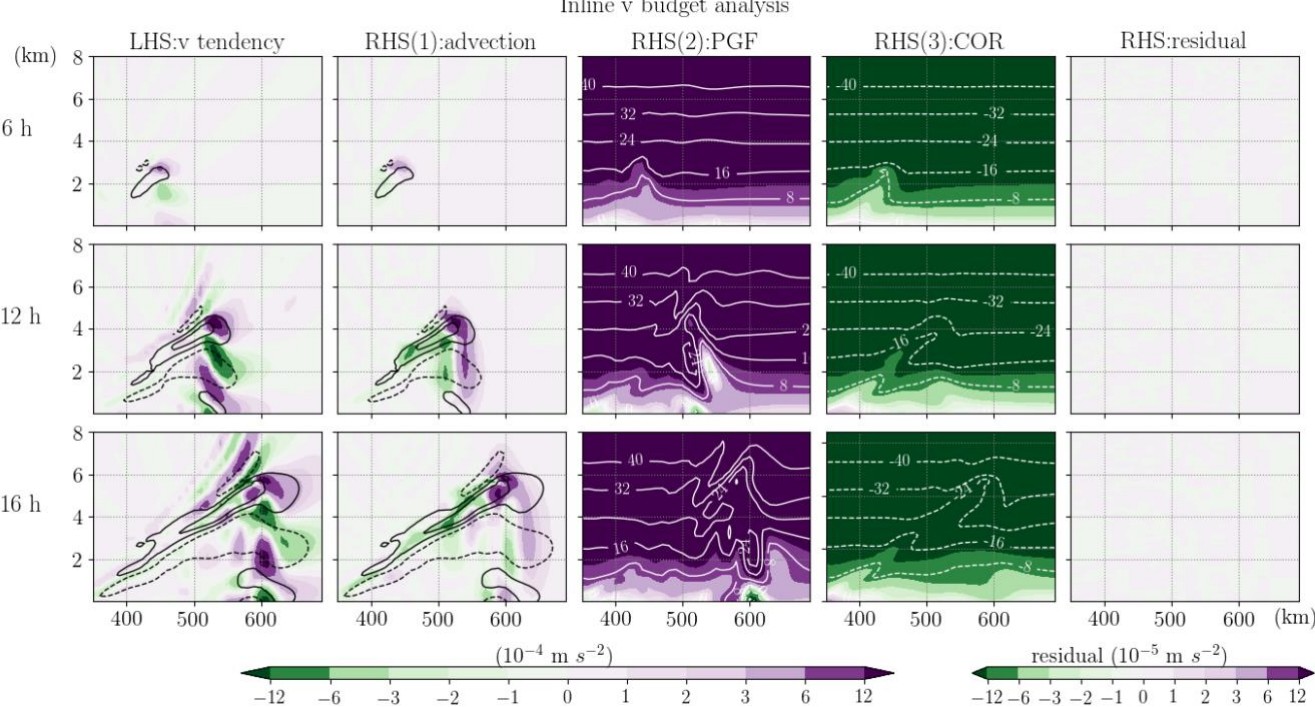

**Figure 3. Inline budget analysis of horizontal momentum, V, with each term extracted directly from the model. In each row, the shaded subplots from the left to right show the term of v tendency, flux-form advection (ADV), horizontal pressure gradient force (PGF), Coriolis force (COR) (white contours indicate the values exceeding the color bar) and residual [Eq. (1); $P_V$ is zero and the generally small curvature term (CUV) is not shown]. All terms are divided by $\mu_d$ and thus have a uniform unit of $ms^{-2}$. The black contours indicated the horizontal velocity v of 2 and 6 $ms^{-1}$ (positive and negative values shown in solid and dashed lines, respectively). Each row from top to bottom illustrates the budget analysis at 6, 12, 16 hour, respectively.**



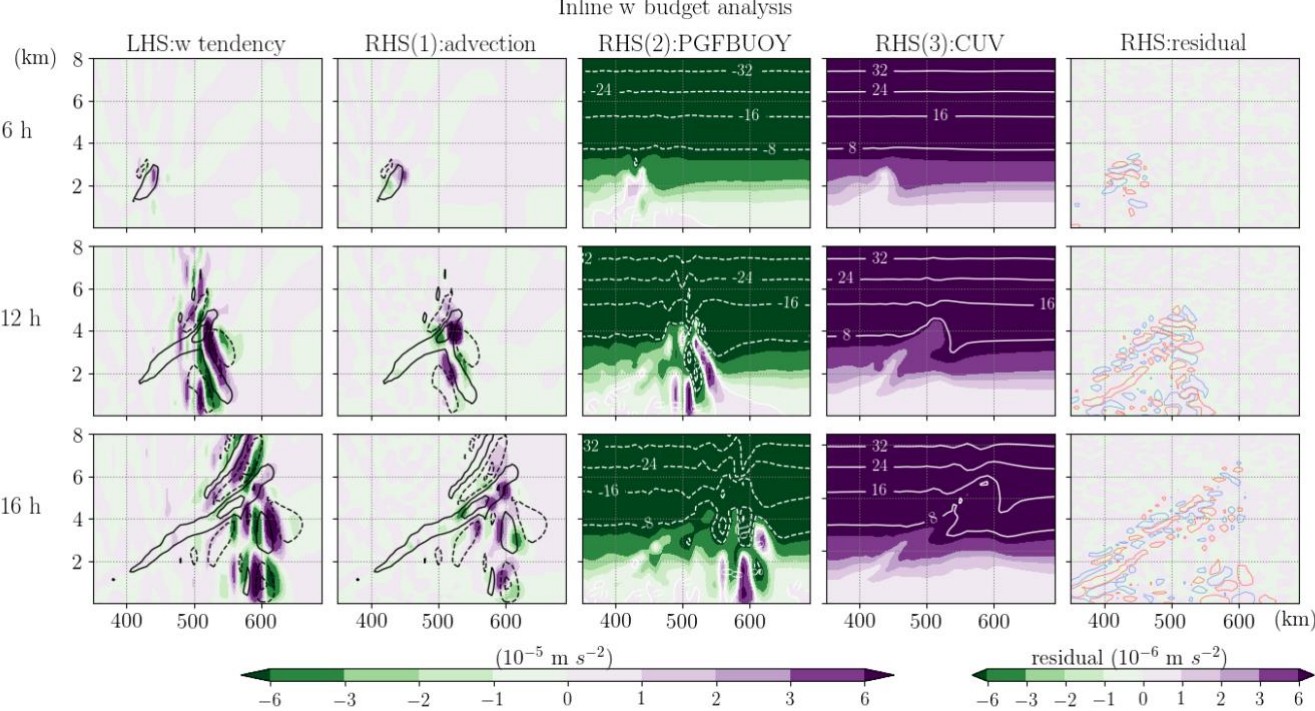

**Figure 4. Inline budget analysis of vertical momentum, W, with each term extracted directly from the model. In each row, the shaded subplots from the left to right show the term of w tendency, advection (ADV), net vertical pressure gradient and buoyancy force (PGFBUOY), curvature (CUV) (white contours indicate the values exceeding the color bar) and residual [Eq. (2); $P_W$ is considered but not shown here]. All terms are divided by $\mu_d$ and thus have a uniform unit of $ms^{-2}$. The black contours indicated the horizontal velocity w of 5 and 15 $cms^{-1}$ (positive and negative values shown in solid and dashed lines, respectively). The red (blue) contours shown in the rightmost column indicate the acoustic-step components of PGFBUOY with a positive (negative) value of $3 \times 10^{-4} \ ms^{-2}$. Each row from top to bottom illustrates the budget analysis at 6, 12, 16 hour, respectively.**



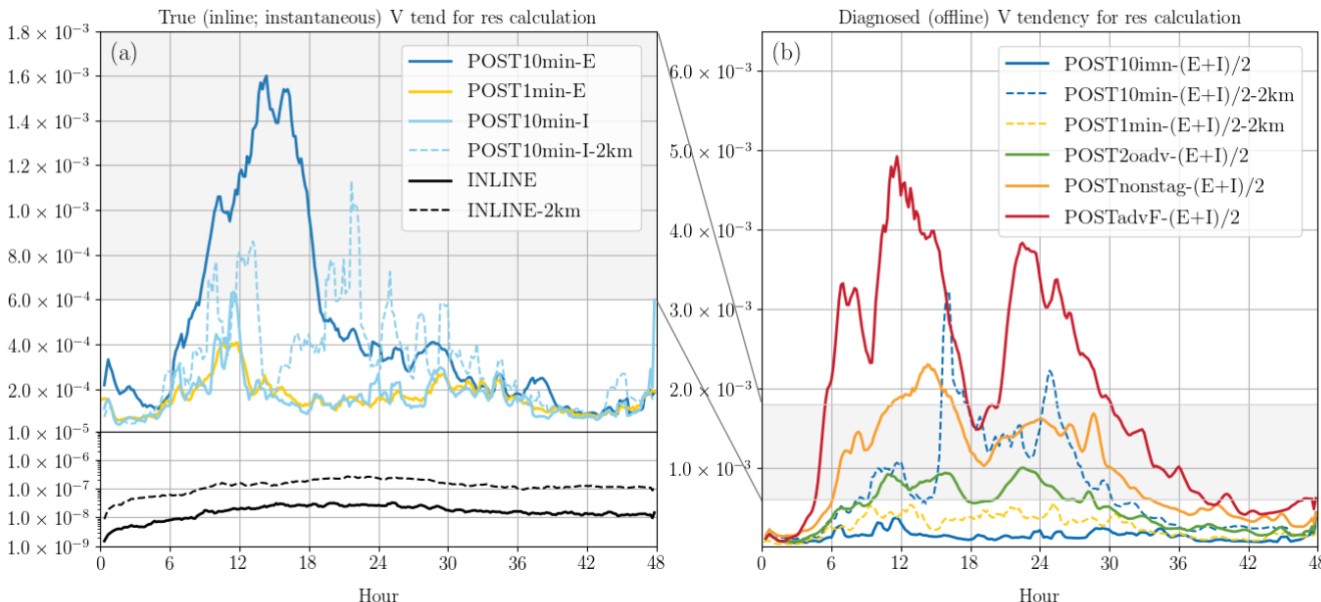

**Figure 5.** Evolution of the magnitude of the 99th percentile residual ($ms^{-2}$) in the horizontal momentum V budget analysis. For the residual calculation, (a) uses the true V tendency (derived during the integration of the model) and (b) uses the post-diagnosed V tendency (Eq. (8) as the lhs term and different colors indicate different post-processed methods for estimating the rhs forcing terms. The residual obtained from the inline budget retrieval are in black. Solid and dashed lines are for the 10-km and 2-km run, respectively.

705



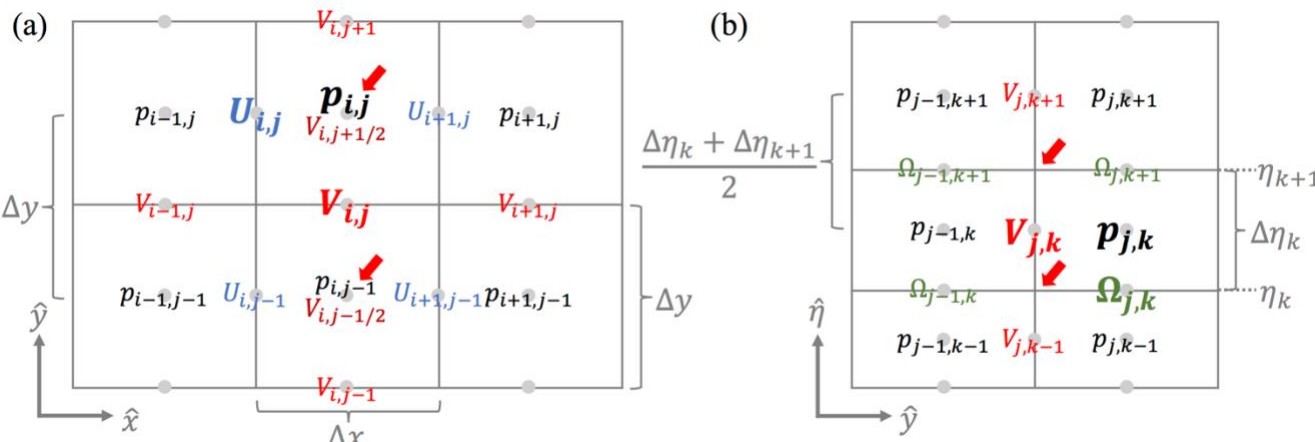

**Figure 6. (a) Horizontal and (b) vertical C staggering grids for different variables in the WRF model. Note that variables ϕ and W are allocated on the same grid as Ω; μ, α and q$_*$ are on grid same as p. The red arrows indicate the grids that would be used to calculate the second-order spatial derivative term for the V momentum at the V-grid (i, j, k).**

710





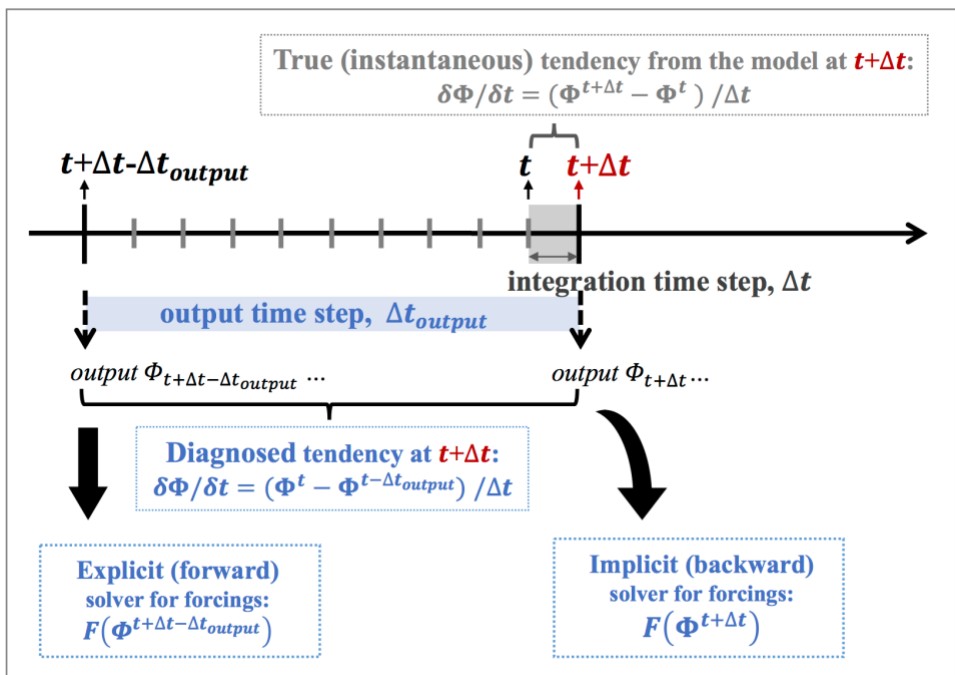

**Figure 7.** Schematic plot showing the explicit (forward) and implicit (backward) solvers for the rhs forcing terms, as well as the diagnosed and the true (calculated inline during the integration of the model) lhs tendency term defined in this study.

715





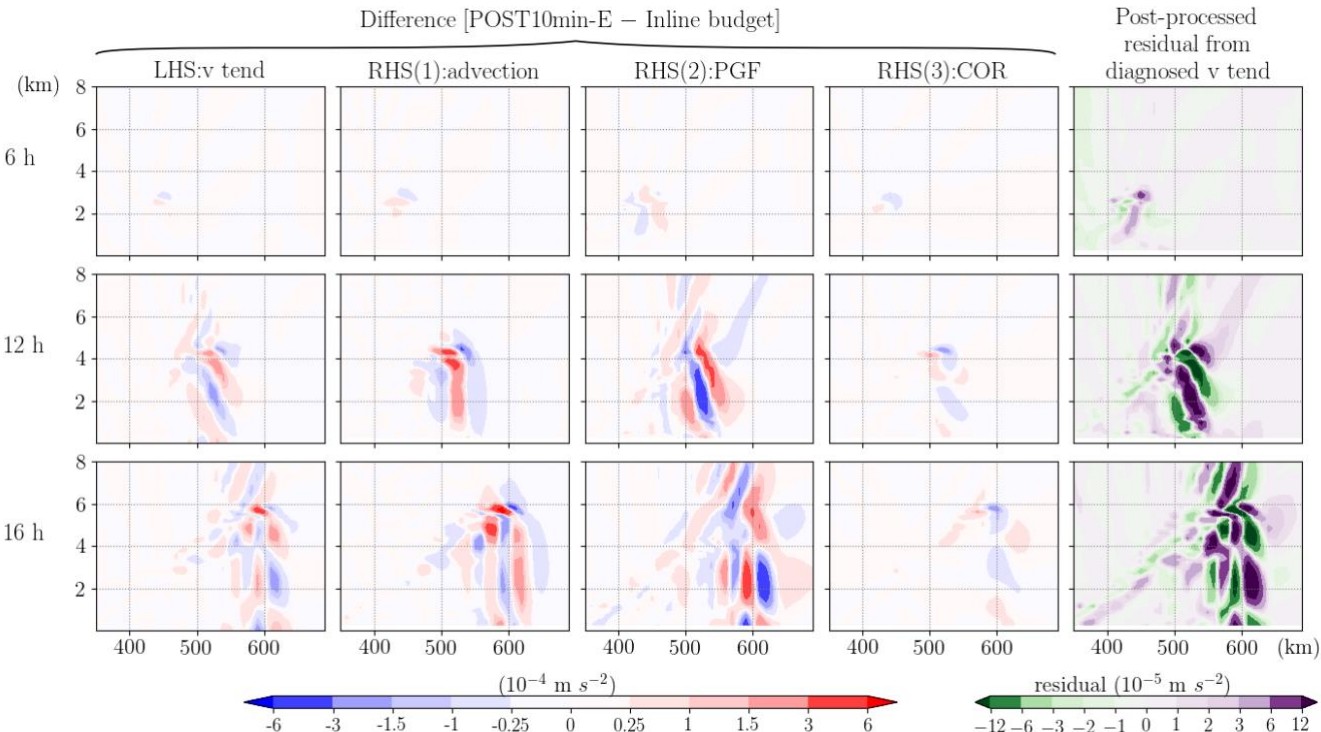

**Figure 8.** The difference between the post-processed (POST10min-E) and inline budget analysis for the horizontal momentum, V. All terms have been divided by $\mu_d$ and thus have a uniform unit of $ms^{-2}$. In each row, from left to right indicates the difference for v tendency, ADV, PGF and COR. The rightmost column indicates the residual term obtained in the post-processed budget analysis. Each row from top to bottom shows the results at 6, 12, and 16 hour, respectively.





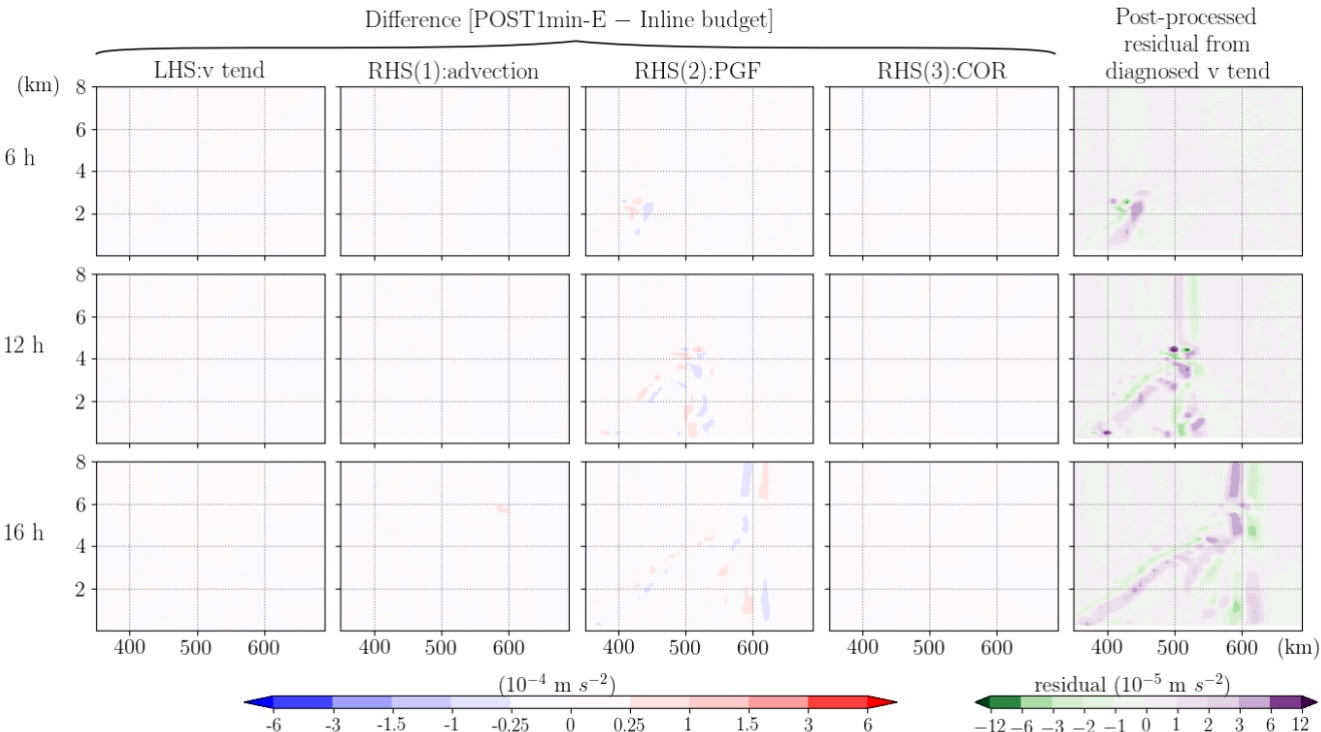

**Figure 9. Same as Fig. 8, but the post-processed budget analysis is applied on the data with an output time interval of 1 minute (POST1min-E).**

725



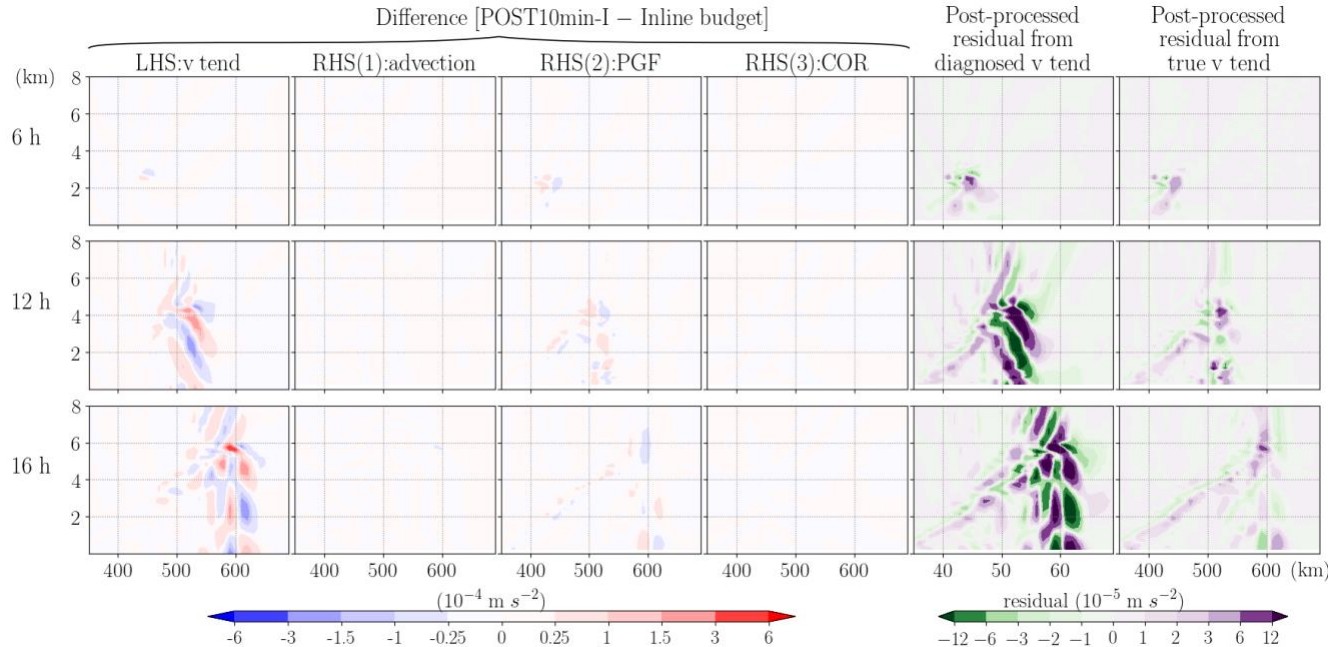

**Figure 10.** Same as Fig. 8, but the post-processed rhs terms are diagnosed implicitly (POST10min-I) and an extra column is added on the rightmost showing the residual from the true tendency (i.e., the instantaneous value obtained from the model).

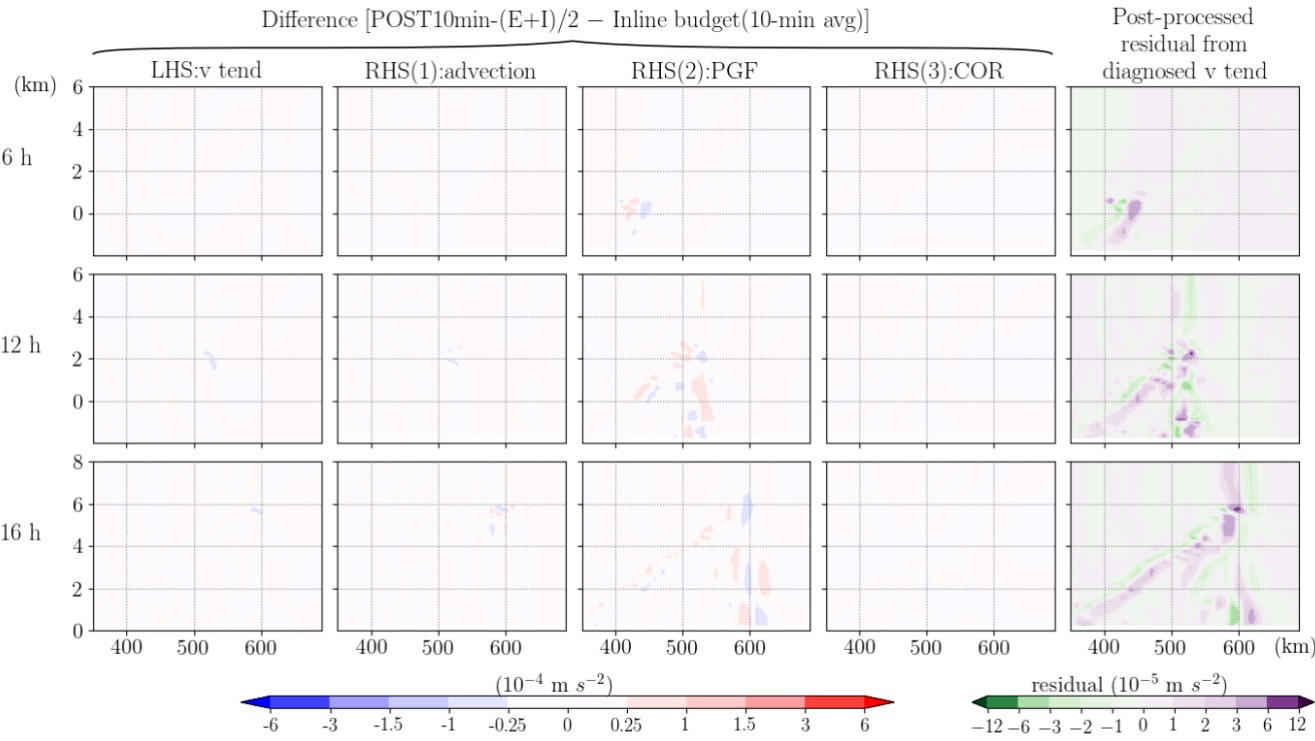

**Figure 11. Same as Fig. 8, but the forcing terms diagnosed in the post-processed budget analysis are the average of explicit and implicit methods (POST10min-(E+I)/2). To represent the same time window as the post-processed analysis, the inline budget results used here for the difference calculation are the 10-min averages (corresponding to the output interval) instead of the instantaneous values.**



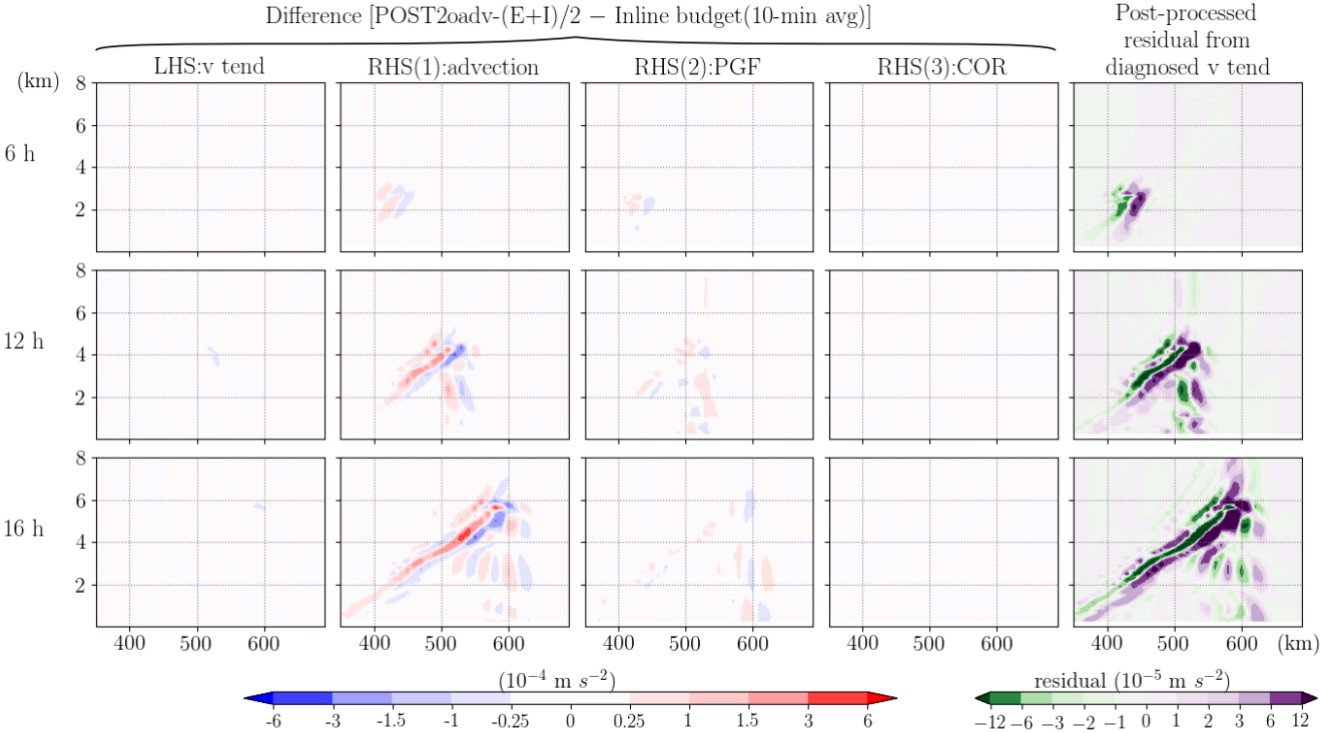

**Figure 12. Same as Fig. 11, but the post-processed analysis uses a second order operators for advection calculation (POST2oadv-(E+I)/2).**



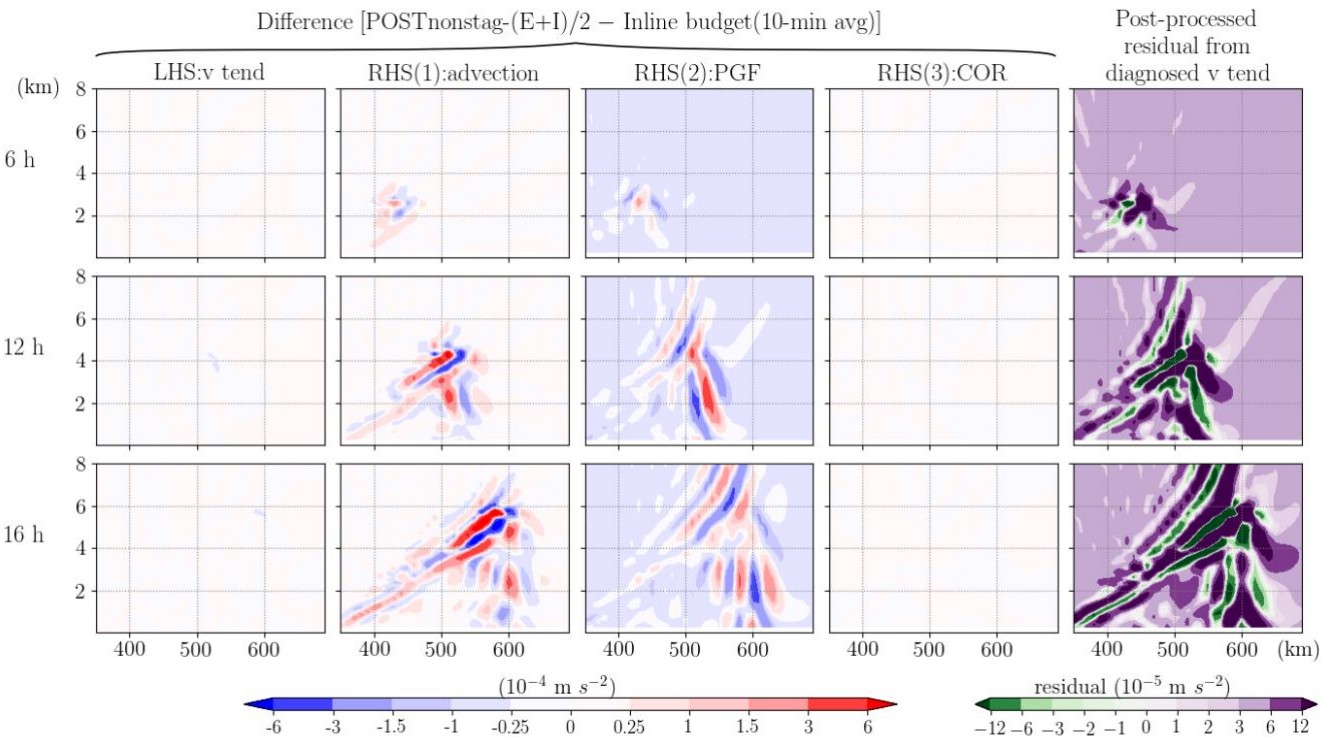

740

**Figure 13. Same as Fig. 11, but for the post-processed analysis does not consider C staggering grids (POSTnonstag-(E+I)/2).**



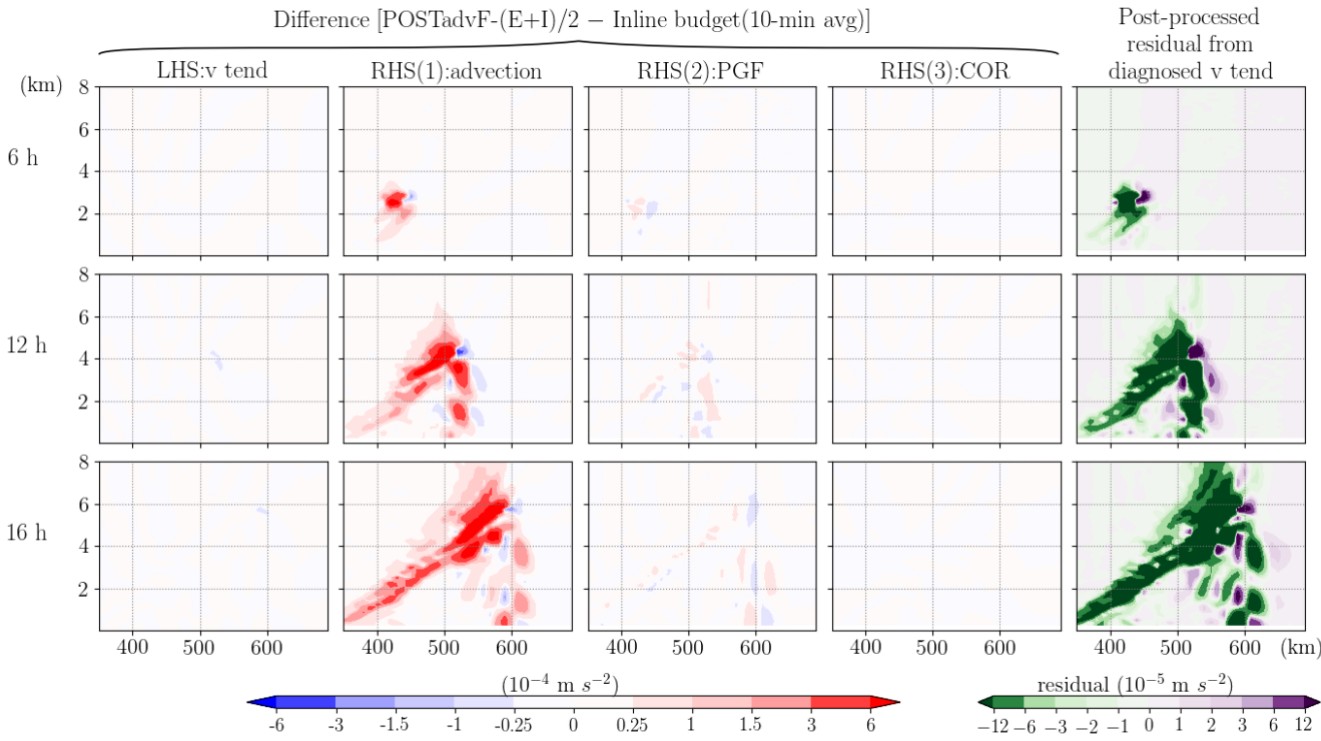

**Figure 14. Same as Fig 11, but for the post-processed analysis is applied on the advective-form equation (POSTadvF-(E+I)/2).**



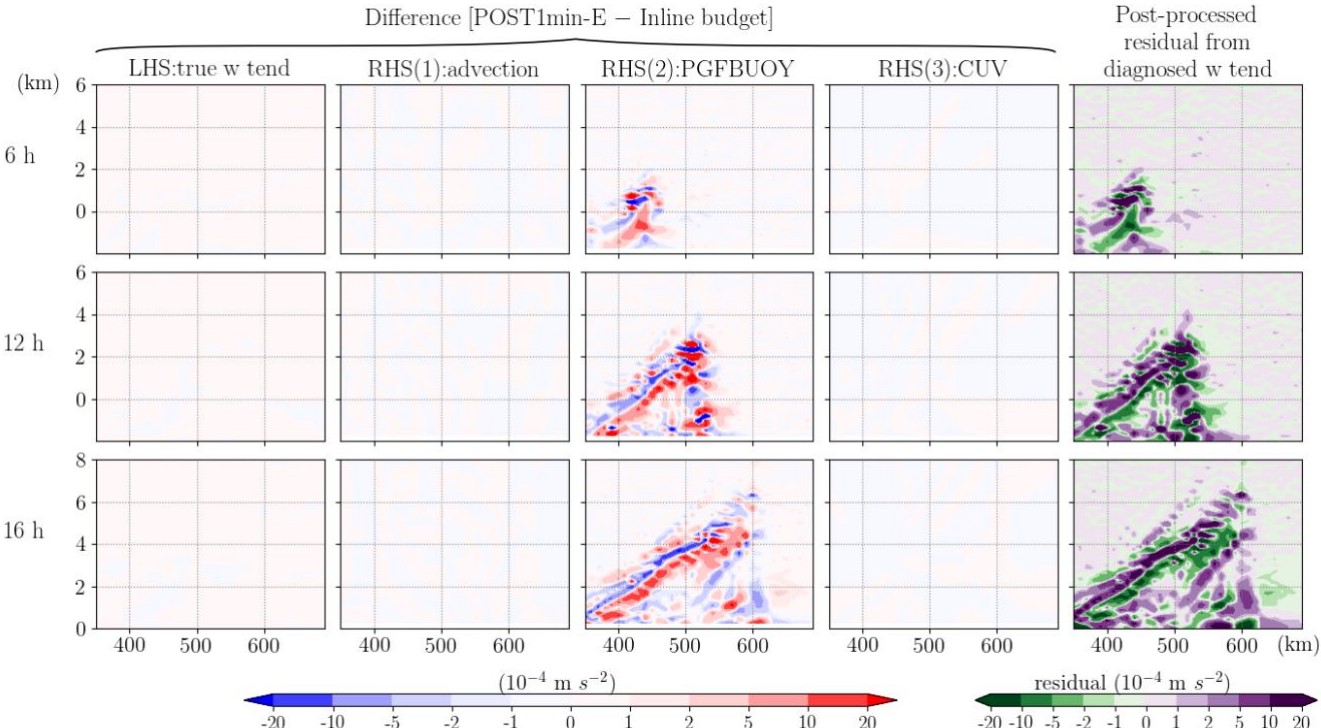

Figure 15. The difference between the post-processed (POST1min-E) and the inline budget analysis for vertical momentum W. All terms have been divided by $\mu_d$ and thus have a uniform unit of $ms^{-2}$. In each row, the subplots from left to right indicates the difference of true W tendency, ADV, PGFBUOY and CUV. The rightmost subplot indicates the residual term obtained in the post-processed budget analysis. Each row from top to bottom shows the results at 6, 12, and 16 hour, respectively.



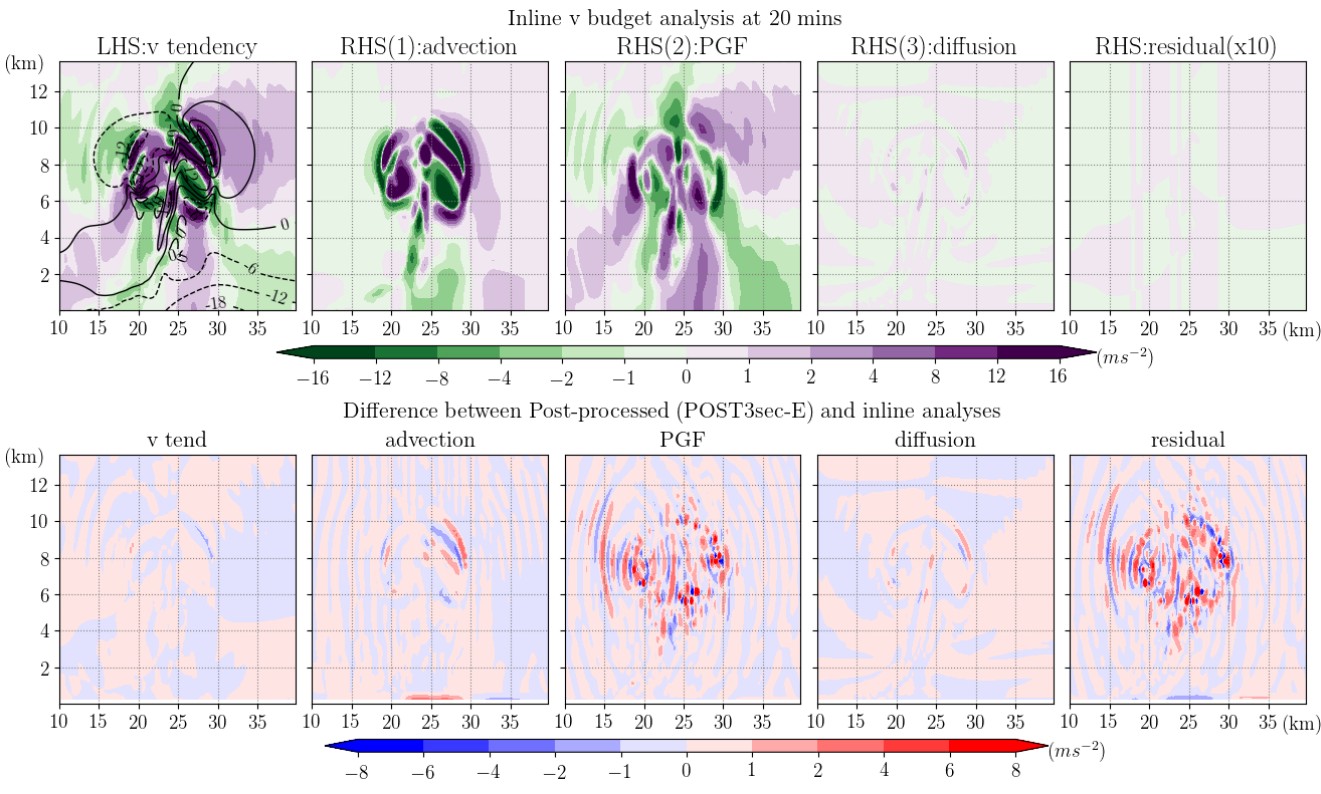

**Figure 16. Upper row shows the inline budget analysis of horizontal momentum, V, for the WRF ideal test case of 2-D squall line at 20 minutes of simulation time. Shading in subplots from left to right represents the term of v tendency, advection (ADV), horizontal pressure gradient force (PGF), diffusion and residual (multiplied by 10 to emphasize its small magnitude as compared to the other terms). The black contours show the velocity, v, with an interval of 6 $ms^{-1}$. The bottom row shows the difference between the post-processed (POST3sec-E) and the inline budget analysis.**