# Peer review of "Towards the closure of momentum budget analyses in the WRF (v3.8.1) model"

_Geoscientific Model Development, 2019_

## Referee Comment (RC1) · Patrick Hawbecker (Referee) · 5 Nov 2019

**General comments**: Towards the closure of momentum budget analyses in the WRF (v3.8.1) model by Chen et al. provides a clear and useful overview of the most common ways to obtain the various budget components from model output. While no novel techniques for calculating the budget components are developed within this study, the utility is within the comparison of existing techniques and highlighting the advantages and shortcomings of these techniques. For a study such as this, the rudimentary 2-dimensional setup is justified as it pertains strictly to the techniques to calculate the budget components and not the physical processes within the simulations. The study focuses analysis on the 99th percentile of the residuals for each method as well as the spatial locations and magnitudes of the individual budget components and residuals at

instances in time throughout the simulation. The authors are careful to draw conclusions that are within the scope of the provided analysis. Overall, the paper is easy to follow and understand, remains on topic, and provides useful guidelines to the community and suggested best practices for calculating budget components moving forward. Further, the explanations of the steps involved in order to repeat such a study appear thorough and the recommended technique for the calculation of budget components is provided via a GitHub repository. As the model utilized is open sourced, it could be beneficial to the community to have this code submitted to be included in a future release.

**Specific comments**: If I am understanding the analysis correctly, it appears that analysis of the residual is confined to analyzing the 99th percentile whereas the figures show both large positive and negative values of the residuals. Should the lower end of the distribution (i.e. the 1st percentile) also be considered here? Further, is there a reason that only the extreme ends of the distribution are considered? If there is no reason, have you performed any sensitivity to the percentile chosen?

**Technical Corrections**:

Figures 3 and 4 are somewhat difficult to see the "canceling out" between PGFBUOY and CUV in the filled contours, but the contour lines do help. Would a log-scaled colorbar be more appropriate here? Or maybe just adding panels to show the difference, PGFBUOY - CUV?

Figure 4 caption: The second to last sentence explaining the rightmost column never mentions the the residual is contoured in the background and reads as though it is plotting strictly the acoustic-step components of PGFBUOY.

Figure 5b legend: blue solid line "POST10imn" should be "POST10min"

Line 313: "Moving the first term on the rhs of Eq. (17) is to the lhs..." should the word "is" be there?

Lines 318 smf 387: Section titled, "3.2.2 Results- horizontal momentum budget" and "3.2.3 Results- vertical momentum budget" should have a space between "Results" and the hyphen

Line 356-358: "Next, we repeated POST10min-(E+I)/2 but applied to the output data that have been interpolated to the universal/un-staggered grid same as the one for the pressure variable (p-grid) (POSTnonstag-(E+I)/2)." I recommend this sentence to be revised. It is not very clear what is going on. Throughout paper and figures: Phrases like "v tendency" sometimes include a capital "v" and other times a lowercase "v" (eg. Figure 2 figure and caption are all lower case, Figure 3 has both lower case and capitalized "v", Figure 5 figure and caption are all capitalized; Lines 207-210 use both lowercase and capital). This should be made consistent throughout the paper.

Line 441: "two time larger" should be "two times larger"

---

## Referee Comment (RC2) · Anonymous Referee #2 · 13 Dec 2019

This paper analyzes methods for horizontal and vertical momentum budget estimation, and compares them to an inline budget retrieval method using the WRF model in a simulated 2-D slantwise convection case. The authors presented clear analysis that point out the shortcomings of using offline budget calculations, and make suggestions to improve the estimation accuracy. The guidance should be useful to the modeling community, and especially to the inexperienced. The paper is generally well written and well organized, and figures are clear. Here are some questions for the authors to consider.

1. Despite the difficulties to obtain a closed budget through an offline method, people are careful using that approach to address the issue at hand. In the example presented in this paper, the slantwise convection simulation, can you say something about how

inaccurate budget calculation could impact your study, and possibly affect the scientific conclusion?

2. It's unfortunate it is concluded that the vertical momentum budget via offline method doesn't work. The discussion may need some work. It is unclear why the acoustic mode would impact the budget since one would expect the larger terms are the vertical PGF, the buoyancy as well as the advection? Supposedly, the acoustic modes are meteorologically insignificant, how do you explain it significantly impact momentum budget? Could it be the nature of vertical motion? It is certainly very variable, and are small scale features itself.

3. This is a minor issue. Would map projection or map-scale factor contribute to the accuracy of momentum budget? I understand it is not relevant in your test case, but map-scale factor will come in in a real data simulation. Should people consider it when doing offline calculations?

4. Would you like to contribute the inline budget code to the WRF repository?

Other minor points:

Lines 166-167: Why do you need the vertical velocity damping option? Using a micro-physics could impact momentum tendency through density variations, and would affect the pressure field too.

Page 9, section 3.2.1(b): The discussion may need to be made simpler and clearer. I was lost. With C-staggering, the computation of PGF for V is easy because pressure is naturally located at 1/DX to the north and south of V grid.

Lines 384-385: What does this sentence mean?

Section 3.2.3 Should vertical diffusion be considered in W budget? Would it account for some of the residual term in Fig 4?

Line 467: Disk space is another big issue.

---

## Short Comment (SC1) · 13 Dec 2019

Dear authors, This paper is very interesting and helpful. I'm giving some suggestions for the paper, as follows:

Line 51: The paper explores and gives details about the computation of the individual terms from the model output and it is mentioned that an inline budget analysis has been reported only in a few studies. Then, it should be important to mention more details about these very few studies which have developed an inline budget analysis, especially the ones that used the WRF model. Did they get the terms directly from the model using the same procedures? The present work also used the same procedure as them?

Line 52: Potter et al. (2018) also used an inline budget analysis in the paper "Dynamical Drivers of the Local Wind Regime in a Himalayan Valley" published in the Journal of Geophysical Research: Atmospheres, and it should be important to cite them. Their paper has also a supplement explaining the WRF code modifications and the modified files.

Line 52: The Thesis of Moisseeva was cited in the paper, but she also published a paper "Dynamical analysis of sea-breeze hodograph rotation in Sardinia" in Atmospheric Chemistry and Physics journal. It is interesting to cite their article instead of the Thesis or to cite both. Their paper has also a supplement explaining the WRF code modifications and the modified files.

Line 193: The authors added the inline calculation for the tendency term outside of the RK3 integration loop, after the microphysics scheme. However, as explained by Lehner (2012) and Moisseeva (2014), ru_tend, rv_tend, and rw_tend are the momentum tendency variables calculated by the WRF model and these variables can be outputted. So, as the tendency term is being calculated by the authors, it should be informed the advantages and reasons for doing it, in comparison to using the tendency variables calculated by the WRF model.

Line 523: Although the authors made available the adapted WRF v3.8.1, it is important to mention which files were modified. It is very helpful to know it since some researchers can compare the default files with the modified files and apply the modifications in other versions of the model, or just to understand in details which modifications were done.

---

## Author Comment (AC1) · 17 Jan 2020

**Corrigendum**

In the previous manuscript that was posted as a discussion paper in the scientific discussion forum of GMD, two figures contain incorrect information:

(1) The original Figure 5 shows the maximum residual magnitude instead of the 99th percentile of the residual magnitude. The corrected Figure 5 appears below and it replaces the original Fig. 5 in the revised manuscript.

[Figure]

Corrected Figure 5. Evolution of the 99th percentile of the residual magnitude ($ms^{-2}$) in the horizontal momentum V budget analysis. For the residual calculation, (a) uses the true V tendency (derived during the integration of the model) and (b) uses the post-diagnosed V tendency (Eq. (8)) as the lhs term. Different colors indicate different post-processed methods for estimating the rhs forcing terms. The residual obtained from the inline budget retrieval are in black. Solid and dashed lines are for the 10-km run and 2-km run, respectively.

(2) The units noted next to the color bars in the original Figure 16 showing "$(ms^{-2})$" are incorrect. They have been corrected to "$(10^{-2} \, ms^{-2})$" in the revised manuscript.

The authors regret any confusion the above errors may have caused. Below we show our responses (marked in blue) to all referee comments (marked in black).

Referee #1 Patrick Hawbecker (hawbecke@ucar.edu)

**General comments:** Towards the closure of momentum budget analyses in the WRF (v3.8.1) model by Chen et al. provides a clear and useful overview of the most common ways to obtain the various budget components from model output. While no novel techniques for calculating the budget components are developed within this study, the utility is within the comparison of existing techniques and highlighting the advantages and shortcomings of these techniques. For a study such as this, the rudimentary 2-dimensional setup is justified as it pertains strictly to the techniques to calculate the budget components and not the physical processes within the simulations. The study focuses analysis on the 99th percentile of the residuals for each method as well as the spatial locations and magnitudes of the individual budget components and residuals at instances in time throughout the simulation. The authors are careful to draw conclusions that are within the scope of the provided analysis. Overall, the paper is easy to follow and understand, remains on topic, and provides useful guidelines to the community and suggested best practices for calculating budget components moving forward. Further, the explanations of the steps involved in order to repeat such a study appear thorough and the recommended technique for the calculation of budget components is provided via a GitHub repository. As the model utilized is open sourced, it could be beneficial to the community to have this code submitted to be included in a future release.

We appreciate the reviewer's positive responses to our work.

**Specific comments:** If I am understanding the analysis correctly, it appears that analysis of the residual is confined to analyzing the 99th percentile whereas the figures show both large positive and negative values of the residuals. Should the lower end of the distribution (i.e. the 1st percentile) also be considered here? Further, is there a reason that only the extreme ends of the distribution are considered? If there is no reason, have you performed any sensitivity to the percentile chosen?

During the process of replying to this comment, we noticed a mistake that the original Figure 5 shows the maximum residual magnitude instead of the $99_{th}$ percentile of the residual magnitude. Figure 5 (see the Figure 5 in the Corrigendum) has now been corrected in the revised manuscript. Note that while the maximum residual shows larger fluctuations with time (with larger magnitude in general) as compared to the $99_{th}$ percentile residual, the general evolutions and the relative

magnitudes among different methods remain unchanged and thus do not affect our conclusions significantly. We should have clarified that the "99th percentile residual" that we intended to show was in terms of the absolute magnitude. Such a choice was made simply because we were interested not only in the distribution of errors but also the peak magnitude regardless of the sign (i.e., the worst-case situation). Considering the maximum magnitude can be easily misled by a single outlier, the 99th percentile was chosen. This issue has been clarified in the revised manuscript (Line 211-213 and Fig. 5 caption): *"To understand how the peak error evolves with time, and to avoid reaching misleading conclusions based on one or more outlying values, the evolution of the 99th percentile magnitude of the residual term is shown."*

To fully address reviewer's concern, we also calculated the 95th percentile residual, which also shows the higher tail of the error distribution. Note that while the relative distribution of some post-processed methods varies with overall smaller magnitudes than their 99th percentiles, our conclusion remains qualitatively similar and the inline retrieval still has the smallest residual. We decided not to include Fig. R1 in the revised manuscript for the sake of simplicity. But the consistent results using the 95th percentile and the maximum residual have been added in the revised manuscript (Line 402-406).

[Figure]

Figure R1. Same as the corrected Figure 5 but for the 95th percentile of the residual magnitude.

**Technical Corrections:**

Figures 3 and 4 are somewhat difficult to see the "canceling out" between PGFBUOY and CUV in the filled contours, but the contour lines do help. Would a log-scaled colorbar be more appropriate here? Or maybe just adding panels to show the difference, PGFBUOY - CUV?

*To address this issue, we have added a column showing the net force of PGF+COR for V and PGFBUOY+CUV for W to Figure 3 and Figure 4, respectively, in the revised manuscript.*

Figure 4 caption: The second to last sentence explaining the rightmost column never mentions the the residual is contoured in the background and reads as though it is plotting strictly the acoustic-step components of PGFBUOY.

*It was mentioned in the beginning of the caption: "In each row, the shaded subplots from the left to right show the term of w tendency, advection (ADV), net vertical pressure gradient and buoyancy force (PGFBUOY), curvature (CUV) (white contours indicate the values exceeding the color bar) and residual". To make it clearer, we have modified the second to last sentence of the caption: "The red (blue) contours shown in the rightmost column, laid on top of the residual (shading), indicate the small-step components of PGFBUOY with a positive (negative) value of..."*

Figure 5b legend: blue solid line "POST10imn" should be "POST10min"

*Corrected. Thanks.*

Line 313: "Moving the first term on the rhs of Eq. (17) is to the lhs…" should the word "is" be there?

*The word is removed (Line 325 in the revised manuscript). Thanks for the correction.*

Lines 318 smf 387: Section titled, "3.2.2 Results- horizontal momentum budget" and "3.2.3 Results- vertical momentum budget" should have a space between "Results" and the hyphen

*The titles have been modified to "3.2.2 Results of horizontal momentum budget" (Line 330) and "3.2.3 Results of vertical momentum budget" (Line 407 in the revised manuscript).*

Line 356-358: "Next, we repeated POST10min-(E+I)/2 but applied to the output data that have been interpolated to the universal/un-staggered grid same as the one for the pressure variable (p-grid) (POSTnonstag-(E+I)/2)." I recommend this sentence to be revised. It is not very clear what is going on. Throughout paper and figures: Phrases like "v tendency" sometimes include a capital "v" and other times a lowercase "v" (eg. Figure 2 figure and caption are all lower case, Figure 3 has both lower case and capitalized "v", Figure 5 figure and caption are all capitalized; Lines 207-210 use both lowercase and capital). This should be made consistent throughout the paper.

The sentence has been modified as follows: *"Next, we repeated POST10min-(E+I)/2 but the calculation is applied after all the variables have been interpolated to the universal/un-staggered grid (pressure grid) (POSTnonstag-(E+I)/2)."* Furthermore, we have clarified and made consistent in the revised paper that, when the lhs tendency term in a coupled momentum budget equation is referred, a capital "V" or "W" is used while the lowercase "v" or "w" tendency indicates the uncoupled momentum calculation (for Fig. 2 only).

In the revised manuscript, some text is added to Line 173-174: *"Figure 2 shows the two-day evolution of the 99th percentiles of v and w (hereafter the lower case indicates that the calculation uses the uncoupled momentum field) and their tendencies."* and in Line 197-198: *"(coupled momentum; hereafter the momentum tendency with capital V or W refers to the lhs term derived for the budget analysis)."*

Line 441: "two time larger" should be "two times larger"

This sentence has been deleted in the revised manuscript.

Anonymous Referee #2

This paper analyzes methods for horizontal and vertical momentum budget estimation, and compares them to an inline budget retrieval method using the WRF model in a simulated 2-D slantwise convection case. The authors presented clear analysis that point out the shortcomings of using offline budget calculations, and make suggestions to improve the estimation accuracy. The guidance should be useful to the modeling community, and especially to the inexperienced. The paper is generally well written and well organized, and figures are clear. Here are some questions for the authors to consider.

We appreciate the reviewer's positive responses to our work.

1. Despite the difficulties to obtain a closed budget through an offline method, people are careful using that approach to address the issue at hand. In the example presented in this paper, the slantwise convection simulation, can you say something about how inaccurate budget calculation could impact your study, and possibly affect the scientific conclusion?

One easy-to-interpret example is POSTadvF-(E+I)/2 as shown in Fig. 14. When such offline budget analysis is done, i.e., following the advective-form momentum equation, the contribution of the advection process to the V tendency is severely overestimated. If the residual is neglected or not shown, authors and/or readers may falsely consider the advection process to be the dominant term governing the evolution of the slantwise updraft. This point has been added in the revised manuscript (Line 388-390).

2. It's unfortunate it is concluded that the vertical momentum budget via offline method doesn't work. The discussion may need some work. It is unclear why the acoustic mode would impact the budget since one would expect the larger terms are the vertical PGF, the buoyancy as well as the advection? Supposedly, the acoustic modes are meteorologically insignificant, how do you explain it significantly impact momentum budget? Could it be the nature of vertical motion? It is certainly very variable, and are small scale features itself.

We realized that the term "acoustic modes" might be misleading as they include not only the sound waves but also partially the gravity waves (Klemp et al. 2007). In the appendix of Klemp et al. (2007), it is mentioned that gravity wave modes are formed due to the designated terms that are

required for sound-wave propagation and *"Consequently, in this vertical coordinate* (i.e., terrain-following hydrostatic pressure coordinate)*, the terms governing the acoustic and gravity wave modes are intermingled to the extent that it does not appear feasible to evaluate any of the gravity wave terms on the large time steps, even if one desired to do so."* In fact, the WRF documentation (Skamarock et al. 2008) specifically mentions that smaller time step is designed to accommodate acoustic and gravity-wave modes. Although the terms "acoustic time step" and "acoustic modes" sometimes misleadingly refer to the small time step in the WRF documentation (Skamarock et al. 2008), we have changed the term "acoustic modes" to "small-step modes" in the revised manuscript to avoid confusion. The above information has been added in the revised manuscript (Line 228-235).

Furthermore, the difficulty of closing the budget for the W equation offline is also related to the fact that the model-solved W equation is implicit with a forward-in-time averaging operator applied on the small-step modes (originally termed as "acoustic modes"):

$$\bar{a}^\tau = \frac{1+\beta}{2} a^{\tau+\Delta\tau} + \frac{1-\beta}{2} a^\tau$$

where $\beta$ is a user-specified parameter. The W budget equation can be written as:

$$W^{\tau+\Delta\tau} - W^\tau = \underbrace{R_W{}^{t^*}}_{large-step\ forcing} + \underbrace{\bar{a}^\tau}_{small-step\ modes}$$

and $\bar{a}^\tau$ contains information of the geopotential variables at the time $\tau + \Delta\tau$ (forecast time), whose tendency solver is coupled with the W tendency equation (see Eqs. (3.11) and (3.12) in Skamarock et al. (2008)). These components are not feasible for an offline calculation. Part of the above discussion was already included in the original manuscript but has been made clearer in the section "3.2.3 Results of vertical momentum budget" in the revised manuscript (starting from Line 407).

We also agree that the budget closure for vertical velocity could be difficult by nature due to its rapid variation in small scales. A closer examination shows that the large-step forcing term, PGFBUOY, varies largely, sometimes with sign changes, from step to step during one integration of the RK3 scheme. This point has also been added in Line 424-430 in the revised manuscript.

3. This is a minor issue. Would map projection or map-scale factor contribute to the accuracy of momentum budget? I understand it is not relevant in your test case, but map-scale factor will come in in a real data simulation. Should people consider it when doing offline calculations?

As a matter of fact, our inline retrieval tool also considers the map projection as the retrieval is coded following the model solver, i.e., the governing equations with map factors included. Physical terms are retrieved when its corresponding subroutine is called. For idealized cases on a Cartesian coordinate, the expression/calculation for the retrieved physical terms remains unchanged but simply that the map factor is 1 during the calculation. Note that we did not alter the original solver but only added codes to retrieve the desired information. For people who wish to conduct an offline calculation for a real case study, the most accurate way would be to follow the original formulas that consider map factors (Skamarock et al.'s (2008) Eqs. (2.23)-(2.25)). This point has been added in the revised manuscript (Line 539-541).

4. Would you like to contribute the inline budget code to the WRF repository?

We appreciate reviewer's suggestion. We would be interested in contributing to the WRF repository although the process requires some more code testing (including software, case and feature testing). While we are confident that the current added code can be similarly applied to the most recently released version of WRF model (v4.1), additional implementing and testing are also required to ensure compatibility.

**Other minor points:**

Lines 166-167: Why do you need the vertical velocity damping option? Using a microphysics could impact momentum tendency through density variations, and would affect the pressure field too.

The vertical velocity damping (Rayleigh damping) is activated for numerical stability when and where the vertical motion approaches the limiting Courant number. Its activation has nothing to do with the purpose of closing the budget analysis. Instead, this shows that the non-physical processes that affect the momentum tendencies (despite small magnitude except over the upper layers) can be considered and outputted via the inline retrieval tool. We agree with your idea that the microphysics can indirectly impact the momentum tendencies. Such information is added in the revised manuscript (Line 165-168):

"*For simplicity, only the Thompson microphysics scheme (Thompson et al. 2008) is used among all the parameterization schemes and the vertical-velocity damping (Skarmarock et al., 2008, chapter 4.5.1) is also activated. The former does not directly contribute to the momentum fields*

*(although it can affect the momentum field indirectly through density and pressure variations) and the latter, contained in $P_W$ in Eqs. (2) and (6), affects only the W momentum budget."*

Page 9, section 3.2.1(b): The discussion may need to be made simpler and clearer. I was lost. With C-staggering, the computation of PGF for V is easy because pressure is naturally located at 1/DX to the north and south of V grid.

The section 3.2.1 is rewritten as follows:

*"...For example, the y-derivative of variable $\Phi$ is calculated using the discrete operator:*

$$\frac{\partial \Phi}{\partial y}_{i,j,k} = \frac{1}{\Delta y}\left(\Phi_{i,j+\frac{1}{2},k} - \Phi_{i,j-\frac{1}{2},k}\right). \tag{9}$$

*The index $(i, j, k)$ corresponds to a location with $(x, y, \eta) = (i\Delta x, j\Delta y, k\Delta \eta)$, where $\Delta x, \Delta y$ and $\Delta \eta$ are the grid lengths in the two horizontal and vertical directions (can be vertically stretched), respectively. The same expression applies for the x- or the $\eta$- derivatives. Grid staggering implies that different variables may be located on different grids, i.e., shifted by a half-grid point from the others as illustrated in Fig. 6. Depending on what variable the spatial derivatives are intended for, Eq. (9) should be carried out on the corresponding grid, which is not necessarily the same as the $\Phi$ grid. For example, for the V tendency, all the associated forcing terms involving the spatial derivatives should be performed on the V grid. More specifically, to calculate the PGF term for the V tendency equation, the term $\frac{\partial p}{\partial y}$ and the term $\frac{\partial p}{\partial \eta}$ in Eq. (1) should be calculated on the V grid but not the pressure grid (p grid). Applying Eq. (9) for $\frac{\partial p}{\partial y}$, the V grid with location indices of $(i, j - \frac{1}{2}, k)$ and $(i, j + \frac{1}{2}, k)$ falls exactly on the p grid and hence no interpolation is required (red arrows in Fig. 6a). However, for $\frac{\partial p}{\partial \eta}$, the pressures on the V grid with indices of $(i, j, k - \frac{1}{2})$ and $(i, j, k + \frac{1}{2})$ must be obtained (red arrows in Fig. 6b) through linear interpolation using their surrounding closest four pressure values, e.g.,*

$$p_{V-grid\left(i,j,k-\frac{1}{2}\right)} = \frac{\frac{1}{2}\left(p_{p-grid(i,j-1,k)} + p_{p-grid(i,j,k)}\right)\frac{\Delta\eta_{k-1}}{2}}{\frac{1}{2}\left(\Delta\eta_k + \Delta\eta_{k-1}\right)} + \frac{\frac{1}{2}\left(p_{p-grid(i,j-1,k-1)} + p_{p-grid(i,j,k-1)}\right)\frac{\Delta\eta_k}{2}}{\frac{1}{2}\left(\Delta\eta_k + \Delta\eta_{k-1}\right)} \tag{10}$$

*which is weighted by the irregular (stretched) vertical grid-lengths (Fig. 6b)..."*

Lines 384-385: What does this sentence mean?

The sentence has been modified as follows: *"Figure 5 also shows that any simplification that is inconsistent with the model solver can severely degrade the accuracy of the post-processed budget analysis."* (Line 399-400 in the revised manuscript).

Section 3.2.3 Should vertical diffusion be considered in W budget? Would it account for some of the residual term in Fig 4?

No explicit diffusion is activated in our idealized setup. The implicit diffusion is embedded in the advection process (ADV in Fig. 4). However, the upper-level vertical velocity damping is activated and indeed considered when calculating the residual. This had been indicated in Line 211-213 in the original manuscript (Line 218-219 in the revised manuscript):

*"While the contribution from the upper-layer vertical velocity damping is not shown in Fig. 4 as it is generally small in the low layers, it is included as part of the rhs ($P_W$) when calculating the residual for the inline budget analysis."*

That said, the w damping is only important over the upper layers and thus not shown in Fig. 4 (also indicated in the figure caption).

Line 467: Disk space is another big issue.

Added (Line 494 in the revised manuscript). Thanks.

Ian Dragaud

iandragaud@lamma.ufrj.br

Dear authors, This paper is very interesting and helpful. I'm giving some suggestions for the paper, as follows:

Line 51: The paper explores and gives details about the computation of the individual terms from the model output and it is mentioned that an inline budget analysis has been reported only in a few studies. Then, it should be important to mention more details about these very few studies which have developed an inline budget analysis, especially the ones that used the WRF model. Did they get the terms directly from the model using the same procedures? The present work also used the same procedure as them?

We appreciate the reviewer's valuable suggestion. We agree that addressing the differences among our work and other inline-budget retrieval works is beneficial. We also appreciate you bringing our attention to other similar works. We have added Appendix A to the revised manuscript to address this issue as follows:

To our knowledge, there are at least three other similar inline budget retrieval works that have been done in the WRF model:

  - *Lehner (2012) with WRF (v3.2.1)*
  - *Moisseeva (2014), Moisseeva and Steyn (2014) with WRF (v3.4.1) – publicly available*
  - *Potter et al. (2018) with WRF (v3.8.1) – publicly available*

- **Lehner (2012)** provides a very detailed instruction of how an inline budget retrieval is done for the WRF model, documented in the appendix of her thesis. The method/code was utilized in a published work studying the mechanisms of the thermally driven cross-basin circulation (Lehner and Whiteman 2014). However, the code was never made publicly available. From the document, it appears that Lehner (2012)'s general procedure of retrieving rhs budget terms during the model integration is essentially the same as our approach, which considers both the large-time-step and small/acoustic-time-step contributions. Furthermore, the individual contribution from different parametrization schemes that are activated in her study was also separately retrieved. While the

general method appears highly similar to our code, the momentum budget retrieval in Lehner (2012) only applies to the horizontal momentum (U and V) whereas our tool includes the budget retrieval for the vertical momentum (W) as well.

- **Moisseeva's (2014) and Moisseeva and Steyn's (2014)** inline budget retrieval tool is also for the horizontal momentum equations, U and V, only. Furthermore, it is simpler than Lehner (2012) as it does not include the acoustic/small-step correction terms. While the large-time-step, non-parameterized terms (e.g., pressure gradient terms, advection, Coriolis terms…) are individually retrieved, Moisseeva and Steyn (2014)'s Registry file only outputs one (summarized) term for all the parameterized physics. And if we understand their code correctly, a small error exists that the parametrized physics term is uncoupled with dry air density whereas the non-parameterized terms are coupled with dry air density.

- **Potter et al. (2018)**'s budget retrieval uses the code adapted from Moisseeva (2014), taking references from Lehner (2012), and is applied to the same version of the WRF model as used in this study (v3.8.1). More components are added from the version used in Moisseeva (2014), including the potential temperature budget, vertical velocity budget, the 6th order diffusion term, parametrized physics term decomposed to boundary and radiation schemes…etc. A major deviation between Potter et al. (2018)'s and our retrieval tools exists in that the acoustic/small-step components are still neglected in Potter et al. (2018). Since Potter et al. (2018) uses the same model version as in our study and their budget retrieval code is publicly available, we downloaded their code and compared the resulted budget analyses (both horizontal and vertical momentum) with ours. A WRF idealized test case of 2D squall line is applied (same case shown in Section 4 in our manuscript).

**V budget analysis**
**Utilizing our budget retrieval code:**

[Figure]

Figure R3-1. Our inline budget analysis of horizontal momentum, V, for the WRF ideal test case of 2-D squall line at 20 minutes of simulation time. Shading in subplots is indicated in the subtitle on the top, including the terms of V tendency, advection, horizontal pressure gradient force (PGF), Coriolis force (COR), curvature (CUV), damping (DAMP2), diffusion (DIFF2) and the residual (multiplied by 10 to emphasize its small magnitude as compared to the other terms). All terms are divided by $\mu_d$ and thus have units of ms$^{-2}$. The black contours show the velocity, v, with an interval of 6 $ms^{-1}$.

**Utilizing Potter et al. (2018)'s code: (**note that a small bug was fixed for parallel computing)

[Figure]

Figure R3-2. Same as Fig. R3-1 but utilizes the budget retrieval code from Potter et al. (2018).

It is no surprise to see the largest difference (yet the relative magnitude to the tendency term seems small) appears in the horizontal pressure gradient term (PGF), as this term contains information contributed by the acoustic/small-time-step integration.

**W budget analysis**
**Utilizing our budget retrieval code:**

[Figure]

Figure R3-3. Same as Fig. R3-1 but for the W budget analysis.

**Utilizing Potter et al. (2018)'s code: (**note that a small bug was fixed for parallel computing)

[Figure]

Figure R3-4. Same as Fig. R3-3 but utilizes the budget retrieval code from Potter et al. (2018).

The residual using Potter et al. (2018)'s tool appears larger for the W budget than that for the V budget. This is consistent with our result that the acoustic/small-step modes are more important in the W budget equation than in the V budget equation and thus ignoring them results in larger errors.

Nevertheless, it is a good sign that all the inline retrieval codes have high similarities in the general procedure even when different model versions are used among the above studies. Thus, we believe that one can easily apply the method described in this study to a newer WRF model version unless its dynamical core is fundamentally changed.

Line 52: Potter et al. (2018) also used an inline budget analysis in the paper "Dynamical Drivers of the Local Wind Regime in a Himalayan Valley" published in the Journal of Geophysical Research: Atmospheres, and it should be important to cite them. Their paper has also a supplement explaining the WRF code modifications and the modified files.

We appreciate the reviewer for providing this useful information. The paper has been cited in the modified manuscript and the discussion shown in the previous comment is added (Line 52, 577-589 in the revised manuscript).

Line 52: The Thesis of Moisseeva was cited in the paper, but she also published a paper "Dynamical analysis of sea-breeze hodograph rotation in Sardinia" in Atmospheric Chemistry and Physics journal. It is interesting to cite their article instead of the Thesis or to cite both. Their paper has also a supplement explaining the WRF code modifications and the modified files.

The published paper (Moisseeva and Steyn 2014) has also been added and discussed in our modified manuscript (Line 52 and Appendix A). Thanks.

**Line 193:** The authors added the inline calculation for the tendency term outside of the RK3 integration loop, after the microphysics scheme. However, as explained by Lehner (2012) and Moisseeva (2014), ru_tend, rv_tend, and rw_tend are the momentum tendency variables calculated by the WRF model and these variables can be outputted. So, as the tendency term is being calculated by the authors, it should be informed the advantages and reasons for doing it, in comparison to using the tendency variables calculated by the WRF model.

Thanks for bringing up this question, as this is a key point that should be clarified. The reason we added the calculation for U, V and W tendencies in the model is to calculate the "true" lhs terms in the budget equations, i.e., the simulated/observed changes of momentum fields with time. Because the purpose of the current study is to evaluate the balance/closure of the budget equations, the lhs term should be estimated independently from the retrieval of all the rhs terms. It is true that the momentum tendency terms, ru_tend, rv_tend ...etc, can be directly outputted from the WRF model by modifying the Registry file as done by Lehner (2012), Moisseeva (2014) and Potter et al. (2018). However, on checking the code, one would realize that these values are only the summation of all the large-time-step rhs forcing terms and do not necessarily represent the real momentum changes considering all the physics. For example, the microphysics, damping, acoustic/small-step modes, etc are not considered. Furthermore, because the ru_tend, rv_tend..etc are essentially the accumulation of all the large-time-step forcings after calling the subroutines for different physical processes (e.g., Coriolis force, advection…), there is no surprise that it would be equal to the summation of all the extracted large-time-step terms (with truncation errors and machine-rounding errors only). But one should note that these model-outputted tendency terms, ru_tend, rv_tend, etc, might deviate from the actual changes in the simulated momentum fields. The above discussion has been added in Line 198-202 and Line 589-592 in the revised manuscript.

Line 523: Although the authors made available the adapted WRF v3.8.1, it is important to mention which files were modified. It is very helpful to know it since some researchers can compare the default files with the modified files and apply the modifications in other versions of the model, or just to understand in details which modifications were done.

We appreciate reviewer's suggestion. The full code of the adapted WRF v3.8.1 on our GitHub repository has tags for the fortran files that have been changed from the defaults. But we agree that it would be beneficial to explicitly write out the changed files in the paper. This information has been added in our modified manuscript (see Line 552-556):

*"In this repository, all the files that remain unchanged from the defaults are tagged as "Initial commit". The modified files for the budget retrieval include the Registry.EM_COMMON within the directory Registry; module_diag_misc.F, module_diagnostic_driver.F and module_physics_addtendc.F within the directory phys; module_after_all_rk_steps.F, module_big_step_utilities_em.F, module_em.F, module_first_rk_step_par2.F, module_small_step_em.F and solve_em.F within the directory dyn_em."*

---

## Author Response (AR2)

Responses to the Referee report

Referee report

The authors have adequately addressed the issues raised previously and hence I recommend the paper to be published after clarifying the minor points listed below.

1. Changing acoustic mode to small-step mode, while clarifying some aspect of what this mode represents, it may lose the real meaning of the term. It may be good to mention that the small-time-deal with both acoustic and gravity wave modes, perhaps in 2.1. I see that you discussed this in 3.2.1, so it is up to you whether you need to add another sentence in earlier sections.

   We have added the information accordingly in Section 2.1 when the small-step mode is first introduced:

   Line 136-138: *"Thus, the perturbation momentum equations to be solved are driven by the large-step forcings and the small-step (sometimes referred as "acoustic-step" although it deals with both acoustic and gravity wave modes (e.g., Klemp et al. 2007, Skamarock et al. 2008)) corrections."*

2. Regarding diffusion I asked last time, do you not use any diff_opt options in the model (other than w_damping and damp_opt)? If so, it should be stated in the paper. Also it is likely that others might want to do the inline budget will have the diffusion on (either through diff_opt or PBL) so a note about this term not being considered in this study may be worth mentioning.

   We didn't use *diff_opt* (*diff_opt*=0) for the slantwise convection cases but we did with the default *diff_opt*=2 for the WRF idealized 2-D squall-line test case (Section 4 in the manuscript). The inline budget tool that has been made public in our GitHub repository (with the tag of GMD_submission1) has retrieval for both *diff_opt*=2 and the diffusion from PBL scheme. We will continuously update the inline retrival tool and to include contributions from a different diffusion option other than *diff_opt*=2 in the future. We have clarified these issues in the revised manuscript:

   Line 165-167: *"For simplicity, the only parameterization used is the Thompson microphysics scheme (Thompson et al. 2008). In addition, the upper-level implicit Rayleigh vertical velocity damping (damp_opt=3) is also activated (Skarmarock et al., 2008, chapter 4.4.2)."*

   Line 169: *"No subgrid turbulence scheme is used (diff_opt=0)."*

   Line 435-437: *"A subgrid turbulence scheme based on the prognostic turbulent kinetic energy equation is activated (diff_opt=2 and km_opt=2; Skamarock et al., 2008, chapter 4.2.4)."*

   Line 559-564: *"The current version includes retrieval for terms of local tendency, advection, horizontal pressure gradient force, net force resulting from vertical pressure gradient and buoyancy, Coriolis force, curvature, upper damping (damp_opt=2 and 3), turbulence/diffusion (diff_opt=2), vertical-velocity damping (w_damping=1) and parameterized physics from the planetary boundary layer scheme (bl_pbl_physics), the radiation scheme (ra_lw_physics and ra_sw_physics), the cumulus scheme (cu_physics), and the shallow cumulus scheme (shcu_physics)."*

Some other changes made in the modified manuscript are all minor corrections for typos, format for the units, italicized variables to be consistent with the ones used in the equations, etc. The texts in figures are also modified correspondingly.

[revised manuscript text omitted]